# Adiabatic and radiative cooling are both important causes of aerosol activation in simulated fog events in Europe

Pratapaditya Ghosh[1,2], Ian Boutle[3], Paul Field[3,4], Adrian Hill[3,5], Marie Mazoyer[6], Katherine J Evans[7], Salil Mahajan[7], Hyun-Gyu Kang[7], Min Xu[7], Wei Zhang[7], and Hamish Gordon[8,2]

[1]Department of Civil and Environmental Engineering, Carnegie Mellon University, 5000 Forbes Avenue, Pittsburgh, 15213, United States
[2]Center for Atmospheric Particle Studies, Carnegie Mellon University, 5000 Forbes Avenue, Pittsburgh, 15213, United States
[3]Met Office, Fitzroy Road, Exeter, EX1 3PB, United Kingdom
[4]School of Earth and Environment, University of Leeds, Leeds, LS2 9JT, United Kingdom
[5]European Center for Medium-Range Weather Forecasting, Reading, UK
[6]CNRM, Université de Toulouse, Météo-France, CNRS, Toulouse, France
[7]Oak Ridge National Laboratory, Oak Ridge, TN, 37831, USA
[8]Department of Chemical Engineering, Carnegie Mellon University, 5000 Forbes Avenue, Pittsburgh, 15213, United States

**Correspondence:** Hamish Gordon (gordon@cmu.edu)

**Abstract.** Aerosol-fog interactions affect the visibility in, and life cycle of, fog and are difficult to represent in weather and climate models. Here we explore processes that impact the simulation of fog droplet number concentrations ($N_d$) at sub-kilometer scale horizontal grid resolutions in the UK Met Office Unified Model. We modify the parameterization of aerosol activation to include droplet activation by radiative cooling in addition to adiabatic cooling, and determine the relative importance of the two cooling mechanisms. We further test the sensitivity of simulated $N_d$ to: (a) interception of droplets by trees and buildings, (b) overestimation of updrafts in temperature inversions, which leads to artificially high $N_d$ values; and (c) potential mechanisms for droplet deactivation due to downward fluctuations in supersaturation.

We evaluate our model against observations from the ParisFog and LANFEX field campaigns, building on evaluation described in the companion paper. Including radiative cooling in the activation mechanism improves how accurately we represent the liquid water path and the vertical structure of the fog in our LANFEX case study. However, with radiative cooling the $N_d$ are overestimated for most of the ParisFog cases, and the LANFEX case. The time-averaged overestimate exceeds a factor of four (the normalized mean bias factor exceeds 3.0) in four out of eleven ParisFog cases. Our sensitivity studies demonstrate how these overestimates can be mitigated. Assuming the overestimate affects both radiative and adiabatic cooling, we find that although radiative cooling is more often the dominant source, both cooling sources can sometimes dominate activation.

## 1 Introduction

Aerosol-cloud and aerosol-fog interactions are complex and often represented in a simplified way in weather and climate models. The effects of aerosols on the optical properties and lifecycle of fog, and resulting effects on Earth's radiative balance, are uncertain. Low visibility due to fog leads to hazardous conditions, especially for transportation, and results in financial

losses (Leung et al., 2020; Kulkarni et al., 2019; Peng et al., 2018; Hao et al., 2017; Abdel-Aty et al., 2011; Gultepe et al., 2007). Lakra and Avishek (2022) provides a broad description of the scientific and social importance of fog. Forecasting the fog lifecycle accurately in numerical weather prediction models is challenging (Boutle et al., 2018; Pithani et al., 2019; Jayakumar et al., 2021; Smith et al., 2021; Kutty et al., 2021; Mazoyer et al., 2022). In the cloud microphysics components of most weather prediction and climate models, aerosols, whether represented prognostically or using climatologies, are usually activated to cloud and fog droplets assuming that adiabatic cooling is the only source of supersaturation. Some models have separate parameterizations to predict visibility (e.g. Clark et al., 2008), but in cases we are aware of the visibility parameterization does not inform other parts of the model, for example the cloud microphysics and radiative transfer schemes.

Radiation fog is characterized by radiative cooling at night, clear skies, and calm conditions. The start of the fog is driven by radiative cooling, and with time the fog layer gradually transitions to a well-mixed fog layer in which adiabatic cooling is also active (Maronga et al., 2015; Boutle et al., 2018). In large eddy simulation studies, and probably in reality, droplets can be activated via both radiative and adiabatic cooling (Poku et al., 2021).

In the companion paper, we evaluated the ability of the UK Met Office Unified Model with interactive double-moment aerosol and cloud microphysics to simulate fog at 500 m grid resolution in case studies from the ParisFog field campaign in 2011. We demonstrated strategies to improve fog droplet number concentrations ($N_d$) while activating the droplets via adiabatic cooling only. While the default model significantly underestimated droplet concentrations, we found that either including an updated version of the aerosol activation parameterization by Abdul-Razzak and Ghan (2000) following Ghosh et al. (2024b), or including a contribution to updraft speeds from sub-grid scale turbulence, could substantially improve model performance. Here, in this paper, we explore the contribution of radiative cooling to activation in the same fog cases, and we additionally examine a single case study from the local and Non-Local Fog Experiment (LANFEX) field campaign in 2014 (Price et al., 2018; Price, 2019) for which we have vertical profiles of fog droplet concentrations.

Several existing simulation studies with bulk aerosol and cloud microphysics schemes include radiative cooling in their activation code. Following Stolaki et al. (2015), Mazoyer et al. (2017) simulated a fog event during the ParisFog field campaign (Haeffelin et al., 2010) using the Meso-NH (Lac et al., 2018) large-eddy simulation (LES) model and found that radiative cooling was the main source of droplet activation. Poku et al. (2021) simulated a fog event during the LANFEX campaign using the Met Office Natural Environment Research Council (NERC) Cloud (MONC) LES model (Brown et al., 2015, 2018), and demonstrated very high contributions of radiative cooling to aerosol activation, sometimes up to 100%. Duconge et al. (2020) and Vié et al. (2024) also simulated the fog vertical structure during LANFEX using the Meso-NH model with radiative cooling included. However, LES simulations cannot readily be used to predict the weather and future climate and are often run in idealized frameworks that only rarely include fully prognostic interactive aerosol and realistic space- and time-varying large-scale forcing via non-periodic boundary conditions. Although there are studies that use numerical weather forecasting (NWP) models, such as Jia et al. (2019), Jayakumar et al. (2021), Yan et al. (2021), and Parde et al. (2022), most of these studies use either a single-moment cloud microphysics scheme in which droplet number concentrations are not prognostic, or they do not include aerosol activation by radiative cooling. An exception is a recent NWP study of fog with aerosol-aware microphysics in the WRF model by Peterka et al. (2024), who found an increase in droplet concentration of around a factor of

three resulted from including activation by radiative cooling. In view of the significant changes found when radiative cooling is included in these simulations, therefore, it is useful to continue to explore including activation by radiative cooling in cloud microphysics schemes used in weather and climate models.

The simple and computationally inexpensive 'ARG' aerosol activation parameterization developed by Abdul-Razzak and Ghan (2000) is very popular in different research models used for both weather and climate, including the WRF-Chem model (Grell et al., 2005; Fast et al., 2006), the Community Earth System Model (CESM, Danabasoglu et al. (2020)), Energy Exascale Earth System Model (E3SM, Golaz et al. (2022)), and the UK Met Office Unified Model in both high-resolution (Field et al., 2023; Gordon et al., 2023) and climate configurations such as the UK Earth System Model (UKESM, Mulcahy et al. (2023)). The ARG scheme was designed for low-level warm clouds, but we showed in the companion paper that it can perform well in fog, though it should ideally be modified in polluted conditions, where kinetic limitations on droplet activation are important. It can also be modified straightforwardly (Peterka et al., 2024) to include radiative cooling, similar to the Shipway and Hill (2012) activation scheme (Poku et al., 2021). However, it is unclear whether including activation by radiative cooling will improve or worsen the scheme's performance in a numerical weather prediction setup that must simulate all types of cloud, including but not limited to fog.

In situations like fog, the resolved updraft speeds are very low. To preserve numerical stability and act as a proxy for non-adiabatic sources of cooling in such cases, most models use a minimum updraft speed, or a minimum width of a distribution of updraft speeds, to activate aerosols in the ARG parameterization (Sullivan et al., 2016; Boutle et al., 2018). A minimum updraft speed of 0.01 $ms^{-1}$ is equivalent to a cooling rate of 0.23 K $hr^{-1}$ assuming a 6.5 K $km^{-1}$ temperature lapse rate. In such simulations, droplet concentrations are determined not by the actual cooling rate but by the minimum updraft speed, which is not a global constant, but varies with time and space as the fog evolves. Only by including radiative cooling explicitly can this spatial and temporal dependence of non-adiabatic activation be represented in models.

Activation is not the only parameterized process in our model, and biases in different parameterizations can lead to compensating errors. We explore the possible effect of other processes on the droplet concentration to some extent here in sensitivity studies, aiming to maintain reasonable good agreement with observations. The interception of fog droplets by trees and buildings could be an important droplet sink (Mazoyer et al., 2017), which is not included by default in NWP models, since cloud microphysics schemes are not designed for fog in general. The entrainment of warm air could result in additional mixing and evaporation of droplets through entrainment-evaporation feedback (Ackerman et al., 2004; Bretherton et al., 2007; Barahona and Nenes, 2007). Fog droplets can undergo collision coalescence or Ostwald ripening (Degefie et al., 2014; Mazoyer et al., 2019, 2022) and form larger and fewer droplets. In specific foggy conditions, Boutle et al. (2018) also found a similar configuration of the model we used overestimated updraft speeds. We explore the influence of these effects on simulated $N_d$.

Several fog campaigns have been designed to understand the microphysical properties during fog formation and development. Following the companion paper, we simulate the fog life cycle during the ParisFog (Haeffelin et al., 2010) field campaign in 2011. In addition, we also simulate a fog event during the LANFEX field campaign in 2014 to study the vertical profile of fog droplets ($N_d$) and the liquid-water path. The LANFEX campaign, an extension from the Cold-Air Pooling Experiment (COLPEX; Price et al. (2011)), was designed to study the life cycle of radiation fog in Bedfordshire and Shropshire in Eng-

land, using field measurements and numerical simulations. This campaign also aimed at understanding the vertical growth of fog layers, and the relative importance of local and non-local processes on radiation fog. In Section 2, we provide an overview of the measurements and the model setup we use in this study. We then proceed to explore various sensitivity analyses. Following this, we evaluate the model's performance in detail. Finally, we discuss the relative importance of adiabatic and radiative cooling processes, and conclude the paper.

## 2 Measurements and Model Description

We use the same set of aerosol and cloud measurements from the 11 radiation and stratus lowering fog events from 15-25 November 2011 during the ParisFog field campaign we describe in the companion paper. In brief, we evaluated our simulations using observations from the Scanning Mobility Particle Sizer (SMPS), the Palas Welas-2020 Particle Counter (WELAS), and the DMT Fog Monitor (FM-100) located at the SIRTA observatory near Paris. We also used surface temperature measurements, radiosonde data, and satellite obervations.

In addition, for this paper we use observations from the the Intensive Observation Period (IOP) 1 of LANFEX (Price et al., 2018), during the night of 24–25 November 2014 at the Met Office observation site in Cardington, Bedfordshire, UK. This nocturnal fog case has been extensively studied, first in a dedicated LES model (Boutle et al., 2018) and then in a model intercomparison study which compared and evaluated LES and single column models (Boutle et al., 2022). Two LES studies have previously used this IOP to study aerosol activation specifically (Poku et al., 2021; Vié et al., 2024). While the intercomparison highlighted progress in modeling the fog onset compared to earlier model intercomparison studies (Bergot et al., 2007), models still deviated substantially from observations of fog properties, reflecting the difficulty of forecasting fog. To evaluate our simulations of the IOP, we use observations from including temperature and relative humidity (RH) profiles from radiosondes, vertical profiles of $N_d$ and LWC from a cloud droplet probe on a tethered balloon, and liquid water path measurements from a radiometer.

We use the same model configuration as described in the companion paper. In summary, we use the UK Met Office Unified Model (UM), where a global model (with a horizontal resolution of $1.87° \times 1.25°$) drives two nested regional domains of resolution 4 km and 500 m centered on the SIRTA observatory (48.713° N, 2.208° E) in Paris (for ParisFog 2011) and the Met Office observational site (52.1015° N, 0.4159° W) at Cardington, UK (for LANFEX 2014). Each regional model domain has $300 \times 300$ grid boxes horizontally, and 70 vertical levels extending to 40 km altitude. We use the RA3 model configuration (Bush et al., 2024) with the bimodal cloud parameterization replaced by the Smith et al (1990) parameterization, as in the companion paper, to ensure sufficient coverage of fog. Our simulations use the double moment aerosol microphysics scheme Global Model of Aerosol Processes (GLOMAP, described by Mann et al. (2010, 2012)) coupled to the double moment cloud microphysics scheme Cloud AeroSol Interacting Microphysics (CASIM, described by Shipway and Hill (2012); Grosvenor et al. (2017); Gordon et al. (2020); Field et al. (2023)) to study fog at high spatial resolution. Our simulations include nitrate aerosols (Jones et al., 2021), in addition to other aerosol species, such as black carbon, organic carbon, sulfate. We use the StratTrop chemistry scheme within the UK Chemistry and Aerosol (UKCA) submodel (Archibald et al., 2020; Gordon et al., 2023) in this study.

This relatively complex chemistry mechanism, which predicts ozone prognostically, is required by the nitrate scheme. As discussed in more detail in Section 3, all simulations use the ARG activation scheme to activate aerosols, using updraft speeds diagnosed at the grid scale with no additional contribution from sub-grid-scale turbulence, but with a minimum updraft speed of 0.01 $\mathrm{ms^{-1}}$.

All results presented in this paper are from our 500 m horizontal resolution model domains for ParisFog and LANFEX. The domain locations are shown on the left panels of Figure 1. The middle panels show the urban surface tiles in the 500 m resolution model, with the measurement sites at the SIRTA observatory and at Cardington as black circles. The right panel shows surface elevation (m); the Seine and Thames river basins are visible in the center and extreme south of the domains, respectively.

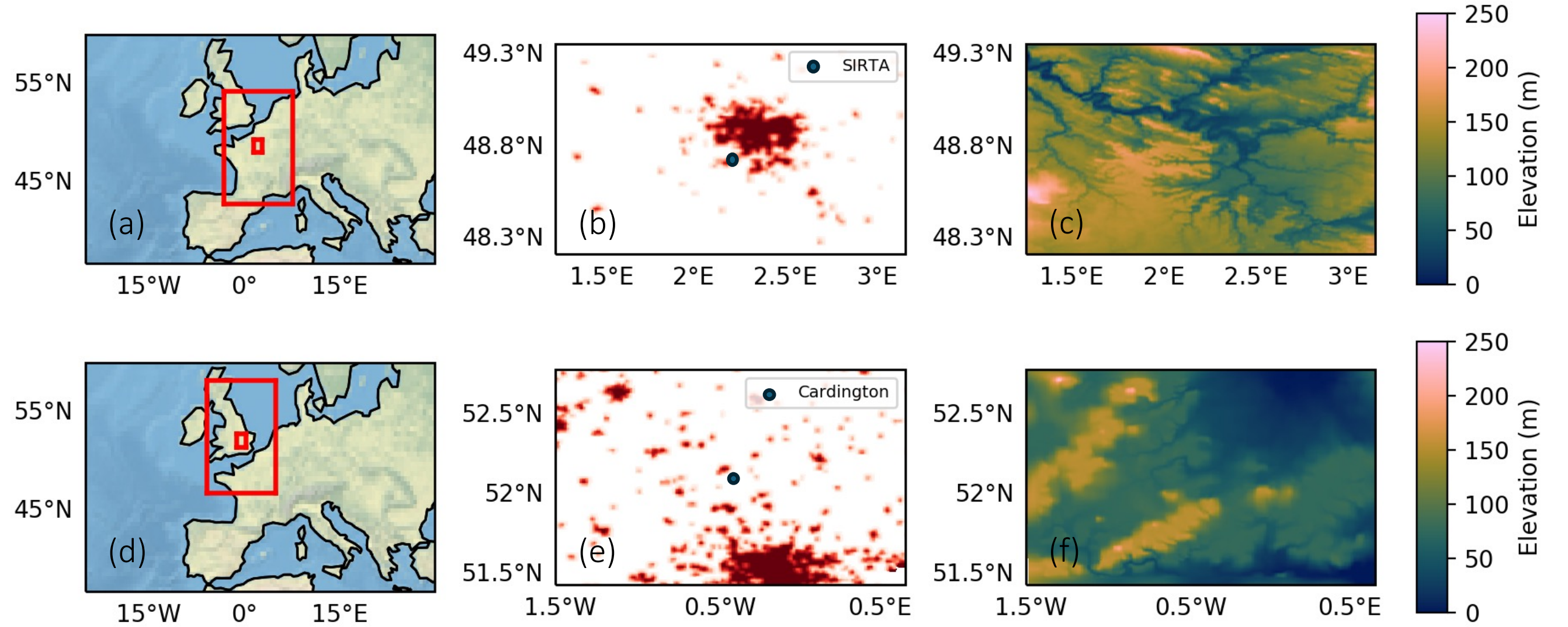

**Figure 1.** The left panels show the two nested domains of grid-spacing 4 km (outer domain) and 500 m (inner domain) used in this study for ParisFog (a) and LANFEX (d). Both domains have 300 grid points in the latitude and longitude directions. The middle panels show urban grid cells in the 500 m resolution model domains for ParisFog (b) and LANFEX (e). The right panels show surface altitude (m) in the same two 500 m resolution domains.

## 3 Aerosol Activation Experiments

In the companion paper, we showed that our model setup for ParisFog with the standard ARG activation scheme significantly underpredicts droplet numbers. The underprediction is likely due to a combination of deficiencies in the ARG activation parameterization and the underestimation of updrafts that results from not including sub-grid turbulence. When we implemented an updated version of the ARG scheme, following the approach of Ghosh et al. (2024b), we were able to predict droplet concentrations reasonably accurately. We additionally updated the aerosol hygroscopicities, though this made relatively little difference to these simulations. Our updated configuration, labelled ‘AD’ for ‘adiabatic cooling’ below, serves as the base-

line simulation in this study. For context, we also show some results here from simulations with the standard ARG activation scheme, labeled Def-ARG. In the companion paper, we demonstrated that accounting for sub-grid turbulence in addition to our updates to the ARG parameterization would lead to overestimation of $N_d$. Given this bias and the uncertainties around how best to parameterize the contribution of sub-grid turbulence to activation, we do not use a sub-grid turbulence component in any of the new simulations we present here. These simulations are described in the following subsections.

### 3.1 Radiative+Adiabatic Cooling (AD-RAD)

Fog usually forms in stable boundary layers. Depending on the balance between mechanical and thermal turbulence, the stable boundary layer can vary from well-mixed to non-turbulent (Stull, 1988). The heat budget inside such a layer is as follows:

$$Q_{\text{total}} = Q_{\text{W}} + Q_{\text{R}} + Q_{\text{E}} + Q_{\text{SH}} + Q_{\text{LH}} + Q_{\text{S}}. \tag{1}$$

The total heat flux $Q_{\text{total}}$ has several components: fluxes from adiabatic cooling $Q_{\text{W}}$ that depend on updrafts, longwave radiative cooling $Q_{\text{R}}$, mixing by entrainment $Q_{\text{E}}$, sensible heat $Q_{\text{SH}}$, latent heat $Q_{\text{LH}}$ and subsidence heating $Q_{\text{S}}$. Updrafts are expected to be low in a stable boundary layer, but sometimes dominate the cooling rate, for example in an 'adiabatic fog' (Boutle et al., 2018). At night, the ground surface cools as it emits infra-red radiation. The cooling rates typically vary between 0.5 K hr$^{-1}$ and 3 K hr$^{-1}$. We assume that adiabatic and radiative cooling are the most important sources of supersaturation in radiation fog. In Meso-NH LES simulations Vié et al. (2024) additionally include surface sensible and latent heat fluxes and turbulent fluxes of temperature while Mazoyer et al. (2017) do not; these terms were not clearly shown to be important, but could be explored in future studies.

In our first sensitivity simulation (termed AD-RAD hereafter), we introduce radiative cooling in the ARG activation scheme following the work of Poku et al. (2021) and Peterka et al. (2024), in addition to changes in simulation 'Mod-Kappa' described in the companion paper (termed 'AD' in this part) of the study (i.e. updated ARG scheme and with modified hygroscopicities). In the def-ARG and AD simulations, the change in supersaturation $\frac{ds}{dt}$ is represented as:

$$\frac{ds}{dt} = \psi_1 \frac{dT}{dt}\bigg|_{\text{ad}} - \gamma \frac{dq}{dt}. \tag{2}$$

In Equation 2, the term $\frac{dT}{dt}\big|_{\text{ad}}$ represents the adiabatic cooling rate, which is given by $-\Gamma w$. Here, $\Gamma = \frac{g}{c_p}$ denotes the adiabatic lapse rate, where $g$ is the gravitational acceleration, $c_p$ is the specific heat capacity of air, and $w$ refers to the updraft velocity. The prefactor $\psi_1$ is defined as $\psi_1 = \frac{c_p}{R_a T} - \frac{L}{R_v T^2}$, where $R_a$ and $R_v$ are the gas constants for dry air and water vapor, respectively, $L$ is the latent heat of vaporization, and $T$ is the ambient air temperature. This term is the the source of supersaturation from adiabatic cooling generated by the updrafts.

The second term is the sink of supersaturation (condensation of water onto aerosol particles and pre-existing droplets), which is dependent on the rate of latent heat release due to condensation of water, $\frac{dq}{dt}$. The prefactor $\gamma$ can be written as: $\gamma = \frac{p}{\epsilon e_{\text{s}}} + \frac{L^2}{R_{\text{v}} T^2 c_{\text{p}}}$. Here, $p$ is the pressure of dry air, and $\epsilon = \frac{R_{\text{a}}}{R_{\text{v}}} = 0.622$.

To introduce radiative cooling, we add an additional source term $\psi_2 \frac{dT}{dt}\Big|_{\text{rad}}$ in the above equation, which now becomes:

$$\frac{ds}{dt} = \psi_1 \frac{dT}{dt}\Big|_{\text{ad}} + \psi_2 \frac{dT}{dt}\Big|_{\text{rad}} - \gamma \frac{dq}{dt}. \tag{3}$$

In Equation 3, the radiative cooling rate is represented by $\frac{dT}{dt}\Big|_{\text{rad}}$ and the prefactor $\psi_2$ is defined as: $\psi_2 = -\frac{L}{R_v T^2}$. We considered both the longwave cooling and shortwave heating predicted by the UM's radiation scheme to calculate the net radiative cooling rate and included it in the activation scheme.

As demonstrated in the 'Results' section, including radiative cooling tends to lead to our simulations overpredicting droplet number concentrations. We explore possible reasons for an overprediction in the remaining sensitivity studies.

### 3.2 Fog Deposition (AD-RAD-SED)

Interception by plant canopies, buildings, and other structures is an important sink of fog droplets (Mazoyer et al., 2017) as they move in turbulent flow near the surface. Sedimentation of cloud droplets is represented in our model, so droplets are lost if they settle out from the lowest model level, but there is no additional parameterization of droplet interception or inertial impaction close to the surface. The LES study of fog by Mazoyer et al. (2017) introduces an additional fog deposition term in the lowest level of their model to account for interception. Taylor et al. (2021) also introduced a similar parameterization for losses due to interception in the WRF model to better simulate LWC and visibility in marine fog.

For our model, the sedimentation (gravitational settling) velocity of a droplet of diameter D, $V_{\text{sed}}$, is (Field et al., 2023):

$$V_{\text{sed}} = a\,D^b \left(\frac{\rho_0}{\rho}\right)^f. \tag{4}$$

In this equation, $a = 3.0\times10^7$ m$^{-1}$ s$^{-1}$, $b = 2.0$, $\rho_0 = 1.2$ kg m$^{-3}$ (reference air density), and $f = 0.50$. From this equation, $V_{\text{sed}} = 0.30$ cm s$^{-1}$ for a 10 µm-diameter droplet and $V_{\text{sed}} = 0.68$ cm s$^{-1}$ for a 15 µm droplet.

To account for interception as well as sedimentation, Mazoyer et al. (2017) used a fixed 2 cm s$^{-1}$ fog droplet deposition velocity ($V_{\text{dep}}$), while Katata (2014) suggested $2.1-8$ cm s$^{-1}$ range of $V_{\text{dep}}$ for short vegetation. Abida et al. (2023) estimated fog deposition velocities to be in the range of 0.08−3.77 cm s$^{-1}$ (mean = 1.95 cm s$^{-1}$) for LWC 0−0.1 µg m$^{-3}$ (typical for our fog cases). Therefore, in this simulation (hereafter termed AD-RAD-SED), in addition to previous changes, we scale up the sedimentation flux (for both the mass and number of cloud droplets) by a factor of 3 in the lowest model level as a proxy for the interception of droplets. From equation 4, the effective deposition velocity is then 0.9 cm s$^{-1}$ for a 10 µm droplet and 2.02 cm s$^{-1}$ for a 15 µm droplet, comparable to these other studies. Although realistic, this crude scaling factor would need refining before being included in a default model configuration.

### 3.3 No activation in inversion (AD-RAD-INV)

In Supplement Figure S1 (right panel), from the 500 m model we show the vertical velocity standard deviation ($\sigma_w$) in the foggy grid boxes for two ParisFog cases and the LANFEX case in the lowest model level (5 m altitude). We find that for the AD-RAD simulation, $\sigma_w$ is about a factor of 5 higher than LANFEX observations.

Radiosonde profiles shown later in Section 4.1, and simulated temperature gradients shown in Figure S2, show that the fog observed during LANFEX IOP1 occurred in a strong temperature inversion. In a temperature inversion, updraft speeds are expected to be close to zero and often negative (see Figure 4 of Stolaki et al. (2015)) because the rising air parcels immediately become cooler than the surrounding air, so they lose their buoyancy. In the model, on average, the resolved updraft in the foggy gridboxes is close to zero, but there are multiple gridboxes where updrafts are positive.

As discussed in the companion paper, in the model, a minimum updraft threshold of 0.01 $\mathrm{ms}^{-1}$ is applied in the activation parameterization, and there is no contribution to updrafts from sub-grid-scale turbulence. As we discuss in the next subsection, the activation scheme updates $N_d$ on each timestep if it exceeds the $N_d$ already present in the grid box, and so the $N_d$ depends on the highest updraft speed since the formation of the fog, likely similar to the maximum shown in the left and middle panels of Figure S1. For the LANFEX case study, fewer than 10% of the resolved updrafts exceed the threshold throughout the fog layer, but for ParisFog updrafts can be greater than 0.01 $\mathrm{ms}^{-1}$ for 20-30% of the gridboxes that have positive updrafts while the temperature is increasing with height as shown in Figure S2. Even though these resolved updraft speeds are usually small, below 0.02 $\mathrm{ms}^{-1}$ at the surface, they do seem to result in significant activation, and it is not clear whether such high resolved updraft speeds within the inversion are physical, or are an artefact of the model's boundary layer scheme.

We hypothesize that activation by adiabatic cooling inside strong temperature inversions should be unlikely, due to weak updrafts and potentially also mixing with subsiding warmer air. In our next sensitivity study, we implement a switch in the activation scheme that suppresses updraft-driven activation when the temperature profile gradient is positive (indicating an inversion). Our simulation, hereafter denoted AD-RAD-INV, incorporates the changes from simulation AD-RAD and this suppression of activation in temperature inversions, which means that aerosols can only activate via radiative cooling when temperature increases with height. Figure 2 shows the resulting decision tree for droplet activation in the AD-RAD-INV simulation.

### 3.4 Droplet Concentration Adjustment Timescale (AD-RAD-DCAT)

In the CASIM cloud microphysics scheme, Equation 5 shows how the droplet number concentration in the model changes from one timestep to the next. The droplet concentration from the previous timestep ('Old $N_d$') is altered by advection and microphysical processes other than activation, such as sedimentation, accretion, or riming, before the activation scheme is called. We call $N_d$ after advection and microphysics but before activation the 'pre-activation $N_d$' and the updated droplet concentration of the current timestep 'New $N_d$'.

$$\text{New}\,N_d = \text{Old}\,N_d + \Delta N_{d,\text{Advection}} + \Delta N_{d,\text{Microphysics}} + \Delta N_{d,\text{Activation}} = \text{Pre-activation}\,N_d + \Delta N_{d,\text{Activation}}. \tag{5}$$

Furthermore, there is the additional constraint that $\Delta N_{d,\text{Activation}}$ can only be positive; if fewer droplets are activated than currently exist, $\Delta N_{d,\text{Activation}}$ is set to zero. If the cloud fraction calculated in the model decreases, the droplets evaporate proportionally to the decrease in the cloud fraction. However, during the entire life cycle of fog, if the fog cover in a gridbox remains the same or increases, there is no mechanism to reduce the droplet concentration in that gridbox except sedimentation (and interception, as included in our study), losses to ice or precipitation, and advection.

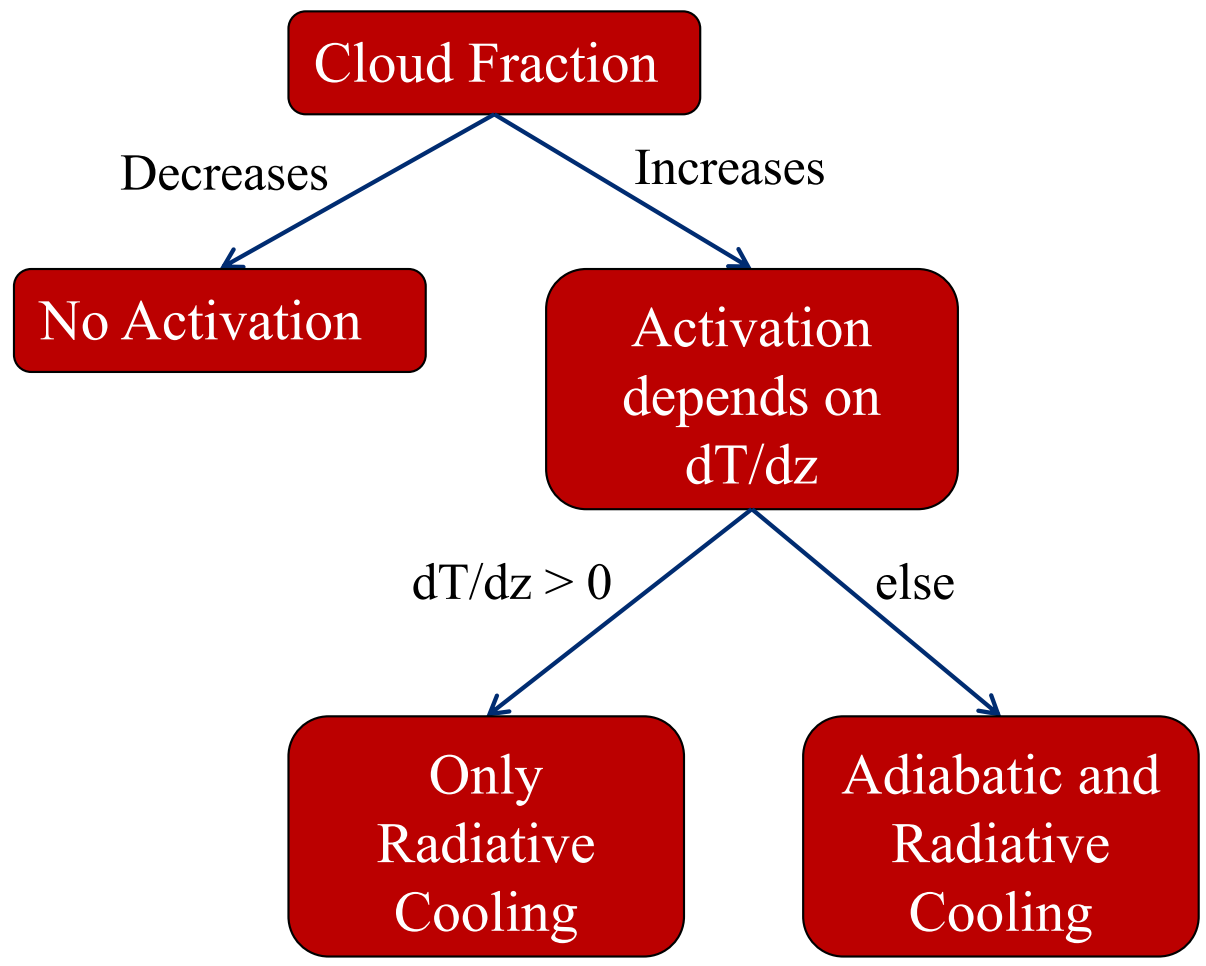

**Figure 2.** This figure shows a flowchart of droplet activation in simulation AD-RAD-INV. The 'ARG' activation scheme is called when cloud fraction increases. Activation is suppressed in temperature inversions. The workflow for simulation AD-RAD is similar, except that activation proceeds independently of dT/dz (the rate of change in temperature with height).

In reality, several other processes may reduce the droplet concentration (or the concentration of droplet-sized particles) during a fog once droplets are formed, but are not included in our model. These include the deactivation of droplets in downward fluctuations in relative humidity (Prabhakaran et al., 2020), collision-coalescence (e.g. Xue et al., 2008; Zhao et al., 2013), Ostwald ripening (e.g. Yang et al., 2018; Mazoyer et al., 2022), or the shrinkage of large, hydrated, but unactivated aerosols that were previously large enough to be detected by a fog monitor or cloud droplet probe. In our model, there are insignificant numbers of unactivated aerosols with an ambient (wet) diameter greater than 2 µm (see the Supplement Figure S2 of the companion paper), yet Mazoyer et al. (2019) find ambient activation diameters in excess of 3 µm (using measurements and $\kappa$-Köhler theory) and significant corresponding 'droplet' concentrations.

We introduce the concept of 'Droplet Concentration Adjustment Timescale (DCAT)' as a proxy for all these other processes that can lead to $N_d$ decreasing over time. We assume that the memory of the maximum supersaturation at which the droplets were activated persists only for a certain time; if the supersaturation decreases, the droplet concentration slowly adjusts to correspond to the concentration that would activate at the subsequent, lower, supersaturations. From the standard equations for droplet growth (e.g. Seinfeld and Pandis, 1998), we calculate that it would take around 700 s for a 15 µm droplet to evaporate to a 1 µm aerosol when the supersaturation changes by 0.05% (relative humidity changes from 100.0% to 99.95%), at 283 K. Small supersaturation changes are likely in fog. Thus, we impose a linearly decaying memory of prior diagnosed supersaturation over a time period of 10 minutes (for our 500 m model, this is 20 timesteps). The choice of 10 minutes is of course somewhat arbitrary, but suffices to examine the sensitivity of $N_d$ to the evolution of supersaturation in the fog. More investigation would be needed to be confident it would generalize from polluted to clean fogs, or to clouds. Thus, in our AD-RAD-DCAT sensitivity simulation, we use activation by adiabatic and radiative cooling as in AD-RAD, together with the condition

**if** (New $N_d >$ Pre-Activation $N_d$) **then**

New $N_d =$ Pre-Activation $N_d + \Delta N_{d,\text{Activation}}$

**else**

New $N_d =$ Pre-Activation $N_d + \frac{\text{dt}}{600} \times \Delta N_{d,\text{Activation}}$

**end if**

### 3.5 Only Radiative Cooling (RAD)

We aim to quantify the relative importance of adiabatic and radiative cooling during the life cycle of fog. By comparing the AD simulation with the AD-RAD simulation, we can quantify the contribution from radiative cooling. However, since these two cooling processes do not result in cloud droplet concentrations that add linearly, in a further simulation labeled 'RAD' we remove the adiabatic cooling term from the source of supersaturation and activate only by radiative cooling. We can then compare this simulation with the AD-RAD simulation to understand the relative importance of adiabatic cooling.

### 3.6 Summary of sensitivity studies

Table 1 shows a brief description of all the simulations used in this study. We aim to provide a direct comparison with the default setup and discuss the importance of aerosol activation via radiative cooling.

**Table 1.** Summary of sensitivity experiments conducted in this study. Here, dT/dz refers to the change in temperature with height. In a temperature inversion, dT/dz is > 0. ARG-New denotes whether ARG parameters are updated, and 'Ad Cool' and 'Rad Cool' whether adiabatic and radiative cooling are included in the simulations.

| Simulation | ARG-New | Ad Cool | Rad Cool | Interception | DCAT |
|---|---|---|---|---|---|
| Def-ARG | No | Yes | No | No | No |
| AD | Yes | Yes | Yes | No | No |
| AD-RAD | Yes | Yes | Yes | No | No |
| AD-RAD-SED | Yes | Yes | Yes | Yes | No |
| AD-RAD-INV | Yes | if dT/dz < 0 | Yes | No | No |
| AD-RAD-DCAT | Yes | Yes | Yes | No | Yes |
| RAD | Yes | No | Yes | No | No |

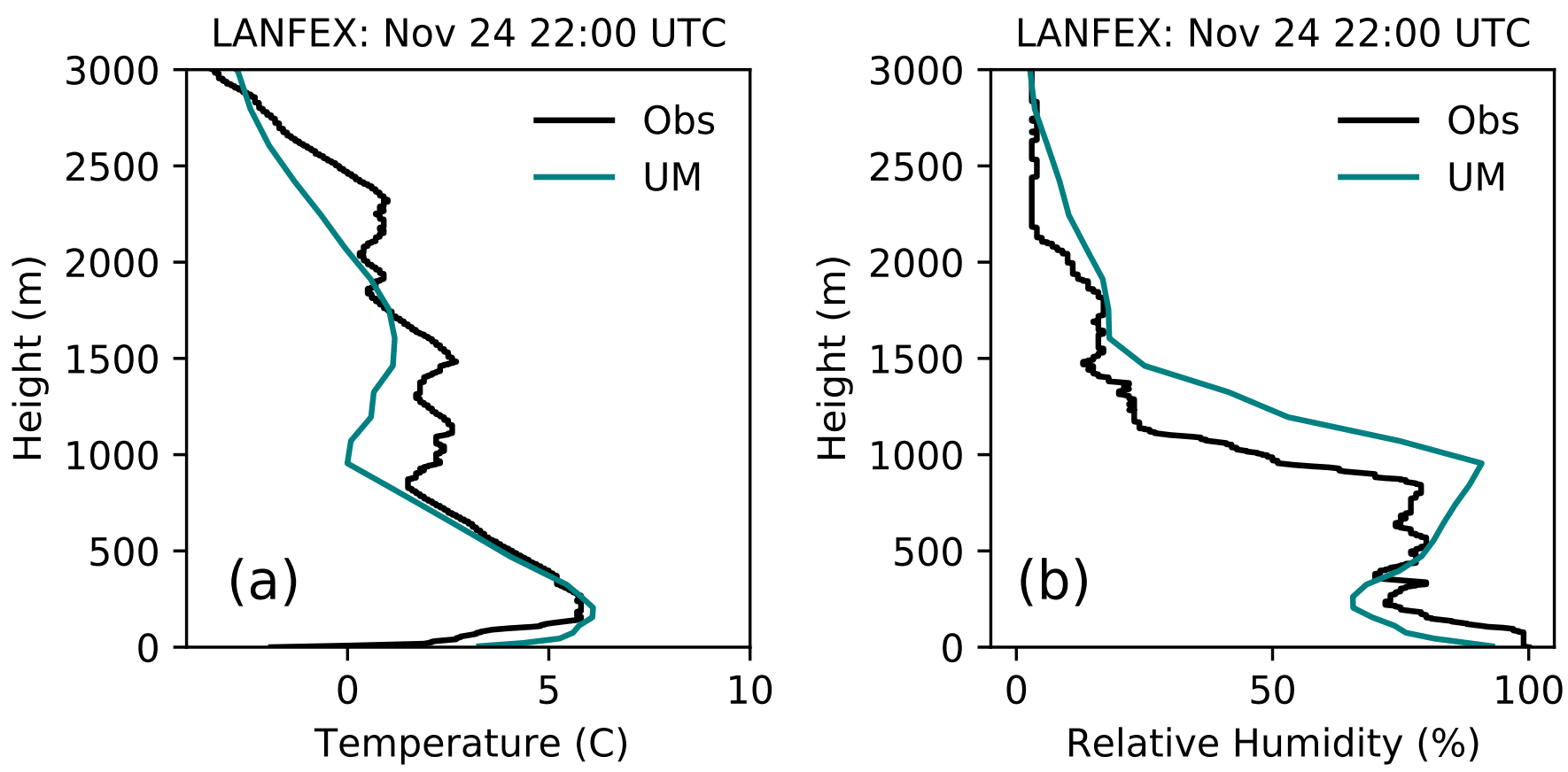

**Figure 3.** Radiosonde profiles of temperature (a) and relative humidity (b) from the LANFEX case study at 22:00 UTC. Model results are from the Def-ARG simulation in our 500 m resolution domain.

# 4 Results

## 4.1 Evaluation of meteorology during LANFEX IOP1

Radiation fog is formed in stable conditions under clear skies. In Figure 3 vertical profiles of temperature and relative humidity (RH) at 22:00 UTC are shown from the LANFEX case study. Similarly to the ParisFog radiosonde profiles shown in the companion paper, the soundings are compared with simulated temperature and RH profile in the 500 m model from the Def-ARG simulation. The fog event is associated with a ground-level inversion, which extends to around 100 m, the top height of the fog in the model and the observations. Simulated temperature and relative humidity near the surface agree well with the observations. The temperature is within 1°and the RH is within 5% of the observations. However, higher in the atmosphere (above 1 km), the agreement is relatively poor, with 2°biases in temperature and 10% in RH. Fortunately, biases above 1 km do not strongly affect the model performance unless they lead to the formation of cloud layers above the fog. Biases of similar magnitude are also found in the ParisFog cases discussed in the companion paper.

For the LANFEX case, the model produces fog roughly at the correct location in our simulations. In Figure 4 we show the spatial variation of the grid average $N_d$ and LWC on 24 November. The model results are near the surface, from the AD simulation, in the 500 m resolution domain. There is sufficient fog in the domain for a statistically robust evaluation of the microphysics. Supplement Figure S3 shows $N_d$ and LWC in simulation AD-RAD, which has slightly greater coverage of fog, and higher $N_d$ and LWC, than simulation AD.

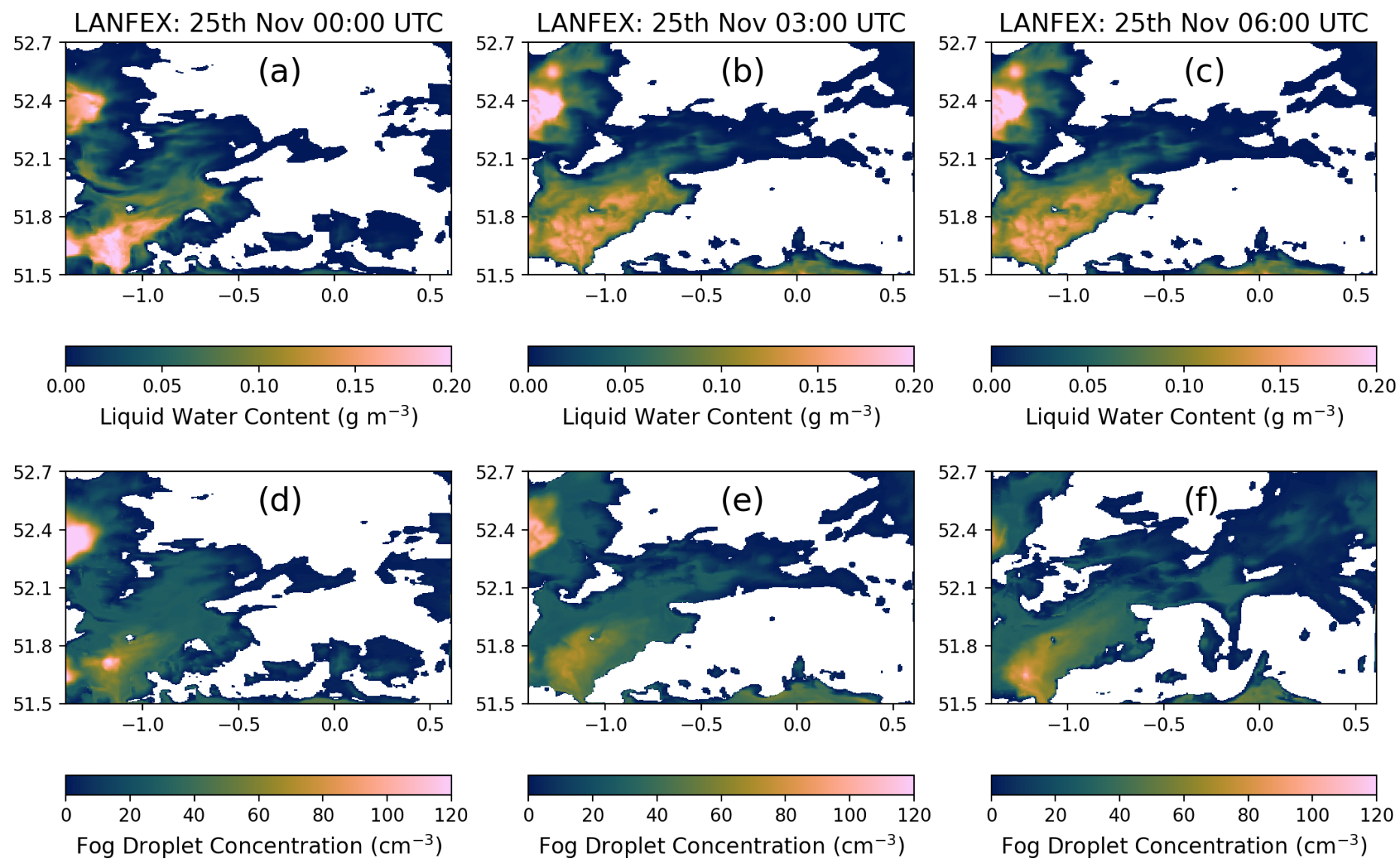

**Figure 4.** Spatial variation of grid-average $N_d$ and LWC on 24 November during the LANFEX field campaign for different times (UTC). We show these properties at 5 m altitude in our 500 m model from simulation AD.

## 4.2 Comparison of size distributions between ParisFog and LANFEX

In Figure 5, we show the dry and ambient aerosol size distribution in the AD simulation in our 500 m model, comparing the ParisFog case on 16 November 2011 with the LANFEX case study for 25 November 2014. The ParisFog observations are from the combination of SMPS, WELAS and fog monitor as described in the companion paper. We also plot the fog droplet size spectrum from the observations during the LANFEX case. Solid lines represent the total number concentrations, while dashed lines represent different aerosol modes and the droplet size distribution. We present an average size distribution from the foggy gridboxes near the surface at 03:00 UTC. Foggy gridboxes are those with at least 20% cloud cover and 0.005 g m$^{-3}$ LWC in that gridbox. We also do not include the 20 gridboxes closest to the edges of the domain. These choices are in line with the companion paper and are maintained throughout the manuscript unless otherwise stated. Unfortunately, observations of aerosol sizes are not available for the LANFEX case study; thus we picked a case from ParisFog that had a similar aerosol size distribution. We also show the $N_d$ distribution from the Def-ARG simulation. The number concentration of aerosol particles greater than 100 nm in diameter at SIRTA during ParisFog is shown in Paper I Figure 8. At the time shown in Figure 5 it is 1700 cm$^{-3}$ while during LANFEX it is 860 cm$^{-3}$. The higher aerosol number concentrations in ParisFog compared to LANFEX are expected given the contrasting urban and rural locations. Coarse mode aerosol number concentrations are one order of magnitude higher in the LANFEX case than in the ParisFog case, but still two orders of magnitude smaller than the simulated droplet concentrations, hence unlikely to impact the simulated fog. The figure also shows that fog droplets, on average, are lower in concentration and larger for LANFEX than for the ParisFog case (about 10 µm compared to 7 µm), as expected given the lower

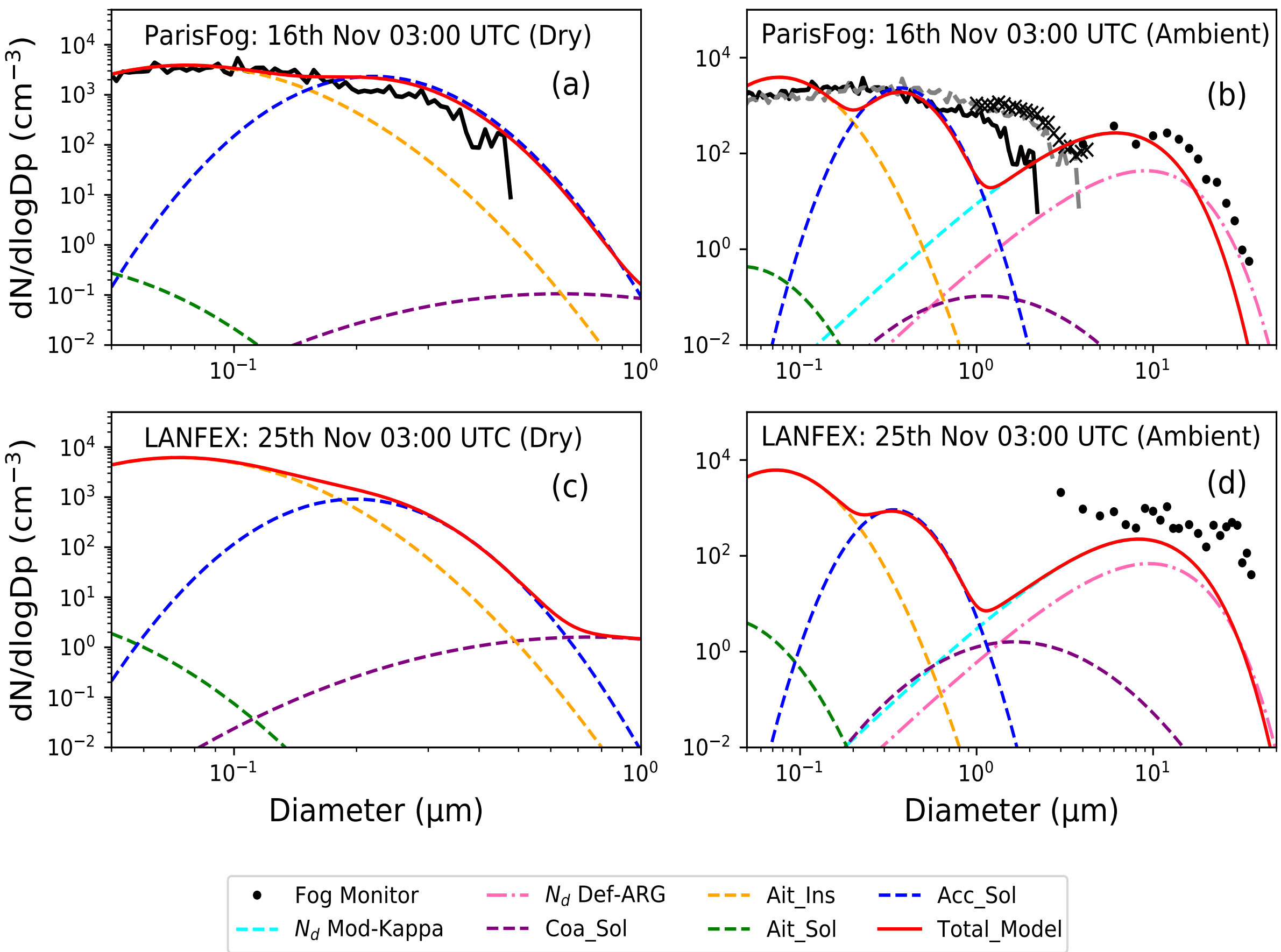

**Figure 5.** Dry and ambient size distribution of aerosols for the 16 Nov 2011 ParisFog case (a, b) and the 25 Nov 2014 LANFEX case study (c, d) from the AD simulation. The average concentrations in different lognormal modes and the gamma distribution of $N_d$ , near the surface of the foggy gridboxes in the 500 m from the AD simulation, are shown at 03:00 UTC for the two cases. Observations are from the SMPS (ParisFog only), the WELAS (ParisFog only), and the fog monitor (ParisFog at 03:00 UTC and LANFEX from a short measurement at 03:30 UTC, compared to hourly simulation output at 03:00 UTC). The $N_d$ distribution from the Def-ARG simulation is also shown using pink dashed line. The aerosol size distributions are almost identical between AD and Def-ARG.

aerosol concentrations. For simulation AD-RAD, similar to the ParisFog case, the droplet concentrations at the surface are in better agreement with the observations.

### 4.3 Timeseries of fog droplet concentration and liquid water content during ParisFog

In Figure 6, we demonstrate the performance of the model in simulating $N_d$ during different ParisFog cases from simulations Def-ARG (green), AD (purple), and AD-RAD (orange). We plot the median (solid and dashed lines) and interquartile ranges (shaded regions) from the foggy gridboxes at the surface. Simulation Def-ARG (shown in green) was designed to understand

the performance of the ARG activation scheme as implemented in the CASIM microphysics code (Gordon et al., 2020; Field et al., 2023) in simulating $N_d$ in fog. We include the updates to the ARG scheme and hygroscopicity, discussed in the companion paper, in simulation AD.

In the AD-RAD simulation (depicted by the orange solid line and shaded regions), radiative cooling is incorporated into the AD simulation as an additional source for generating supersaturation during droplet activation. We calculate the Normalized Mean Bias Factor (NMBF) and the Normalized Mean Error Factor (NMEF), defined as

$$\mathrm{NMBF}\,(\overline{M} \geq \overline{O}) = \frac{\sum_i (M_i - O_i)}{\sum_i O_i}$$

$$\mathrm{NMBF}\,(\overline{M} < \overline{O}) = \frac{\sum_i (M_i - O_i)}{\sum_i M_i}$$

$$\mathrm{NMEF}\,(\overline{M} \geq \overline{O}) = \frac{\sum_i |M_i - O_i|}{\sum_i O_i}$$

$$\mathrm{NMEF}\,(\overline{M} < \overline{O}) = \frac{\sum_i |M_i - O_i|}{\sum_i M_i}$$

Here, $M_i$ are the model data, $O_i$ is the observation data, $\overline{M}$ is the model mean, and $\overline{O}$ is the observation mean. NMBF has a range of $-\infty$ to $+\infty$, and NMEF has a range of 0 to $+\infty$. We report NMBF and NMEF for all fog cases in Supplement Tables S1 and S2 and use them as a tool to compare the model performance among different simulations.

As shown in Paper I, simulation AD realistically simulates $N_d$ for all fog cases, within a factor of 2 of the observations most of the time, while simulation Def-ARG usually underestimates them. As expected, in simulation AD-RAD, the $N_d$ are higher than in simulation AD because of droplet activation via radiative cooling. The difference in $N_d$ is about 50-100 cm$^{-3}$ for all fog events except those on 16$^{th}$ November. For the fog cases on 16$^{th}$ November, radiative cooling is unimportant and simulations AD and AD-RAD behave similarly. Otherwise, the model bias increases for all the fog cases. For example, on 15$_{th}$, 18$_{th}$ and 24$_{th}$ November (first case) (subfigure 6a, 6d and 6i), AD-RAD tends to overestimate $N_d$, and the NMBF (NMEF) change from 2.71, 0.82, 0.23 (2.76, 1.03, 0.81) in AD to 3.59, 3.88, 1.69 (3.64, 3.88, 2.12) in AD-RAD.

In Figure 7, we show timeseries of $N_d$ for ParisFog cases for our sensitivity studies: AD-RAD-SED, AD-RAD-INV, and AD-RAD-DCAT. We show median (solid and dashed lines) and interquartile range (shaded region) from the foggy gridboxes at the surface in the 500 m model. We also show the median $N_d$ from the AD-RAD simulation for ease of comparison. We apply a LWC threshold of 0.005 gm$^{-3}$ and a cloud fraction threshold of 20% to define a grid box as ‘foggy’, and the fog event is defined as the period for which over 1000 grid boxes in the model domain are ‘foggy’.

In simulation AD-RAD-SED (red), increasing the sedimentation rate at the lowest model level to represent the interception of fog droplets by trees and buildings substantially reduces the droplet number concentrations (compared to AD-RAD) during all fog events. In this simulation, the overestimation of $N_d$ in AD-RAD is reduced, and the model agrees better with the observations. For the 16$^{th}$ November cases, the model is now biased low (the NMBF is -0.70 and -0.51). For other fog cases, the biases have been reduced compared to AD-RAD. For example, for the first fog case on 24$^{th}$ November, the NMEF is now 0.60 compared to 2.12 in AD-RAD. Compared to simulation AD, performance improvement is also found in six fog

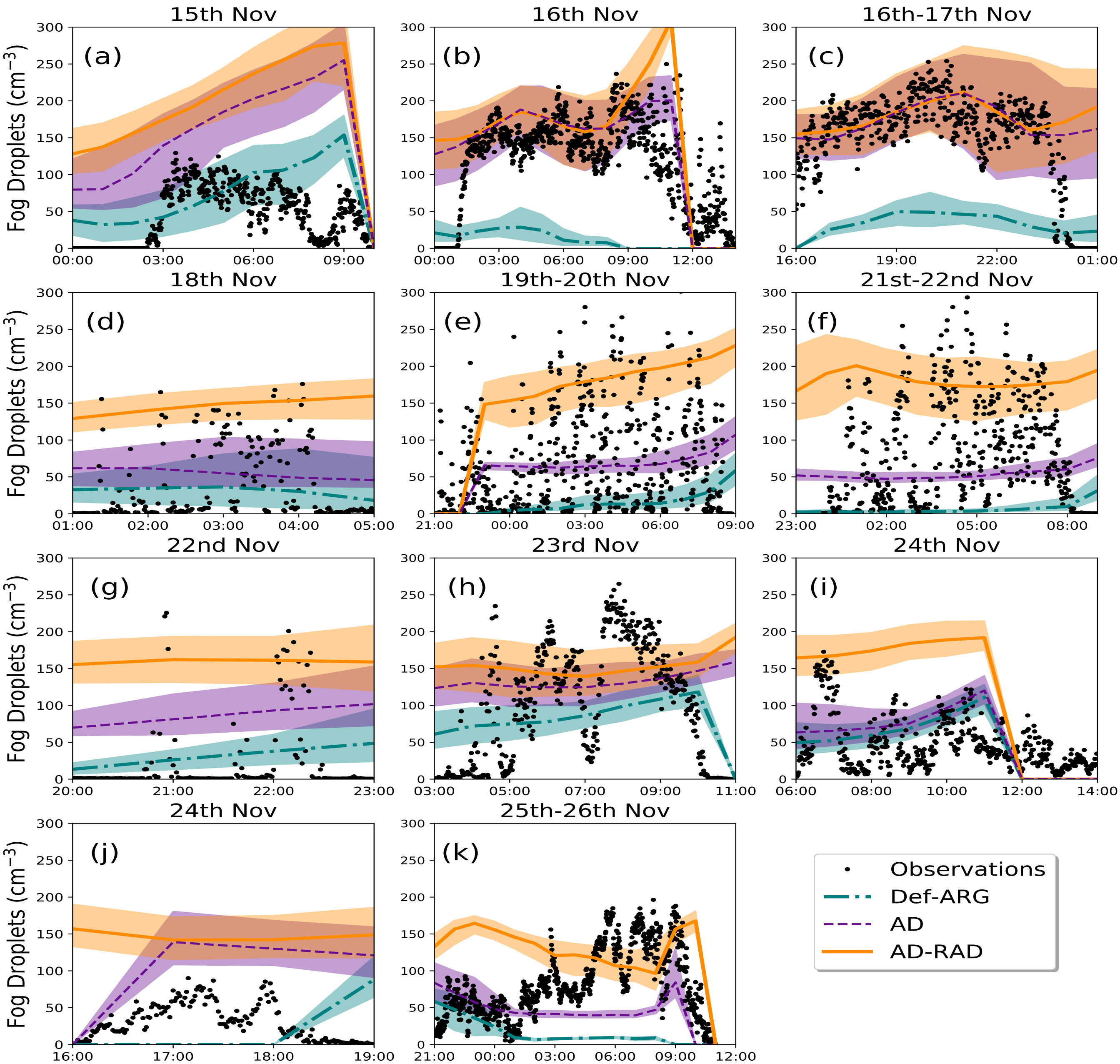

**Figure 6.** Variation of simulated and observed fog droplet concentration as a function of time for different fog events. The results of the 500 m model at 5 m altitude from the simulations Def-ARG, AD, and AD-RAD are compared with observations at the SIRTA observatory (UTC time). Lines represent the median value and the shaded regions represent the interquartile range over the foggy gridboxes.

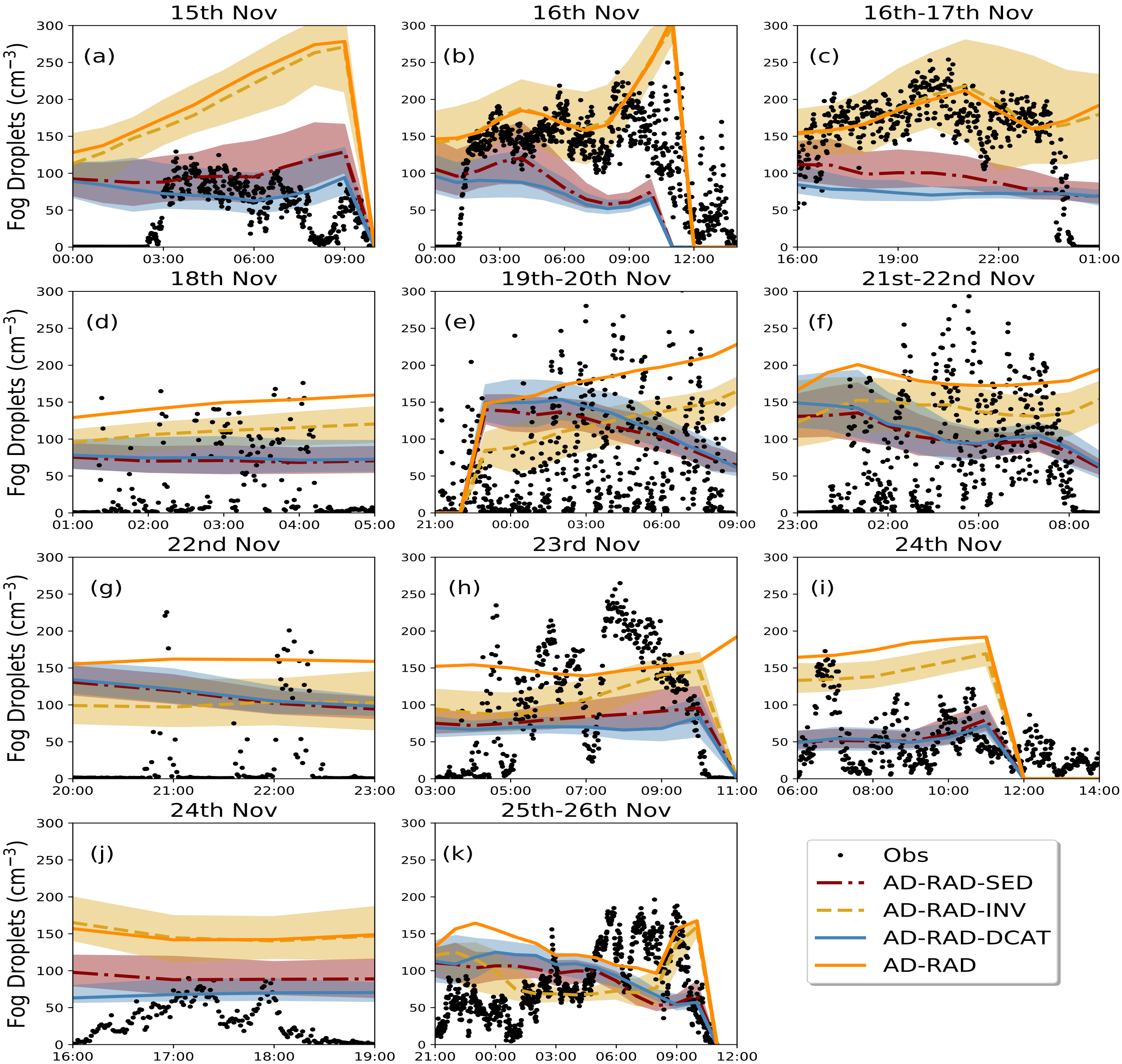

**Figure 7.** Variation of simulated and observed fog droplet concentration as a function of time for different fog events. The median and interquartile range from the 500 m model at 5 m altitude from the simulations AD-RAD, AD-RAD-SED, AD-RAD-INV, and AD-RAD-DCAT are compared with observations at the SIRTA observatory (UTC time). Lines represent the median value and the shaded regions represent the interquartile range over the foggy gridboxes.

cases out of 11. For example, on 15th, 19th, and 26th November cases, the model is in better agreement with the observation in simulation AD-RAD-SED, compared to simulation AD. The NMBF (NMEF) changes from 2.71, -0.42, -0.67 (2.76, 1.01, 0.97) to 1.24, 0.11, 0.11 (1.29, 0.86, 0.63). This simulation supports the findings of other studies (e.g. Mazoyer et al., 2017) that fog deposition is an important physical process that affects the droplet budget, although we cannot be sure that our crude modification accurately simulates real losses due to interception.

In simulation AD-RAD-INV (yellow), we prevent activation in temperature inversions (in addition to changes in AD-RAD). Similarly to the AD-RAD-SED simulation, we find that simulation AD-RAD-INV is also a significant improvement over the AD-RAD simulation, though not (for the majority of cases) over the AD simulation. Our hypothesis leads to minimal changes (compare the yellow solid lines with the dotted orange lines) in $N_d$ for fog cases on 15th, 16th (both events) and 24th November (second event). However, for other fog cases, the changes in $N_d$ are significant. Compared to simulation AD-RAD, 10 out of 11 fog cases have lower biases.

The AD-RAD-DCAT simulation (blue) introduces the droplet concentration adjustment timescale, testing the sensitivity to several physical processes and model artifacts that affect the droplet budget in the fog. This simulation performs similar to AD-RAD-SED and improves the model performance compared to AD-RAD for most cases, and compared to AD for a slim majority of cases (6 out of 11). For example, on 15th, 18th and 24th November, the model now agrees well with the observation compared to AD-RAD. However, on 16 November, there is a factor of 2 underestimation. This test suggests that improving the representation of physical processes like collision-coalescence, or the hygroscopic growth of unactivated haze aerosols, may be important to improve fog simulation in climate and weather models.

There are other possible explanations for the overestimation of droplet concentrations when radiative cooling is included in the aerosol activation mechanism we do not test. For example, Boutle et al. (2018) suggested that in their UKV simulation, which is similar to ours but without interactive aerosols, their radiative cooling rates could be too high. There may also be a positive feedback mechanism that could exacerbate the problem (Boutle et al., 2018): more, smaller droplets can absorb radiation from the surface more efficiently leading to more radiative cooling, which may then result in more activation.

In Figure 8, we compare the timeseries of in-fog liquid water content in foggy gridboxes (LWC) with observations. Subplots (a-k) denote different ParisFog cases. We show median and interquartile ranges (using lines and shaded regions) from the foggy gridboxes at the surface in our Def-ARG, AD and AD-RAD simulations. In-fog liquid water content is calculated by dividing the grid average LWC by the cloud cover. Since low-LWC grid boxes also tend to have a low cloud fraction, the minimum LWC in the timeseries is the LWC threshold for a box to be 'foggy' of 0.005 $\mathrm{g\,m^{-3}}$ divided by the cloud fraction threshold of 20%. The inset histograms of in-fog LWC show the variability between grid boxes at a representative time during each fog event, from simulation AD. On November 22 and 23, the LWC in the AD-RAD simulation closely resembles that of the Def-ARG and AD simulations. As in the companion paper and as expected from simulations of liquid water path evaluated in other studies (Boutle et al., 2022), the model has relatively little skill in representing the trends in LWC during the observed fog events, and the AD simulation shown in the companion paper clearly performs better than AD-RAD. Generally, LWC tends to increase when radiative cooling is included in aerosol activation in the AD-RAD simulation, except during the first fog event on November 16 (subfigure b). LWC is often over-predicted during the early stages of the fog, for example on 15th the 21st and

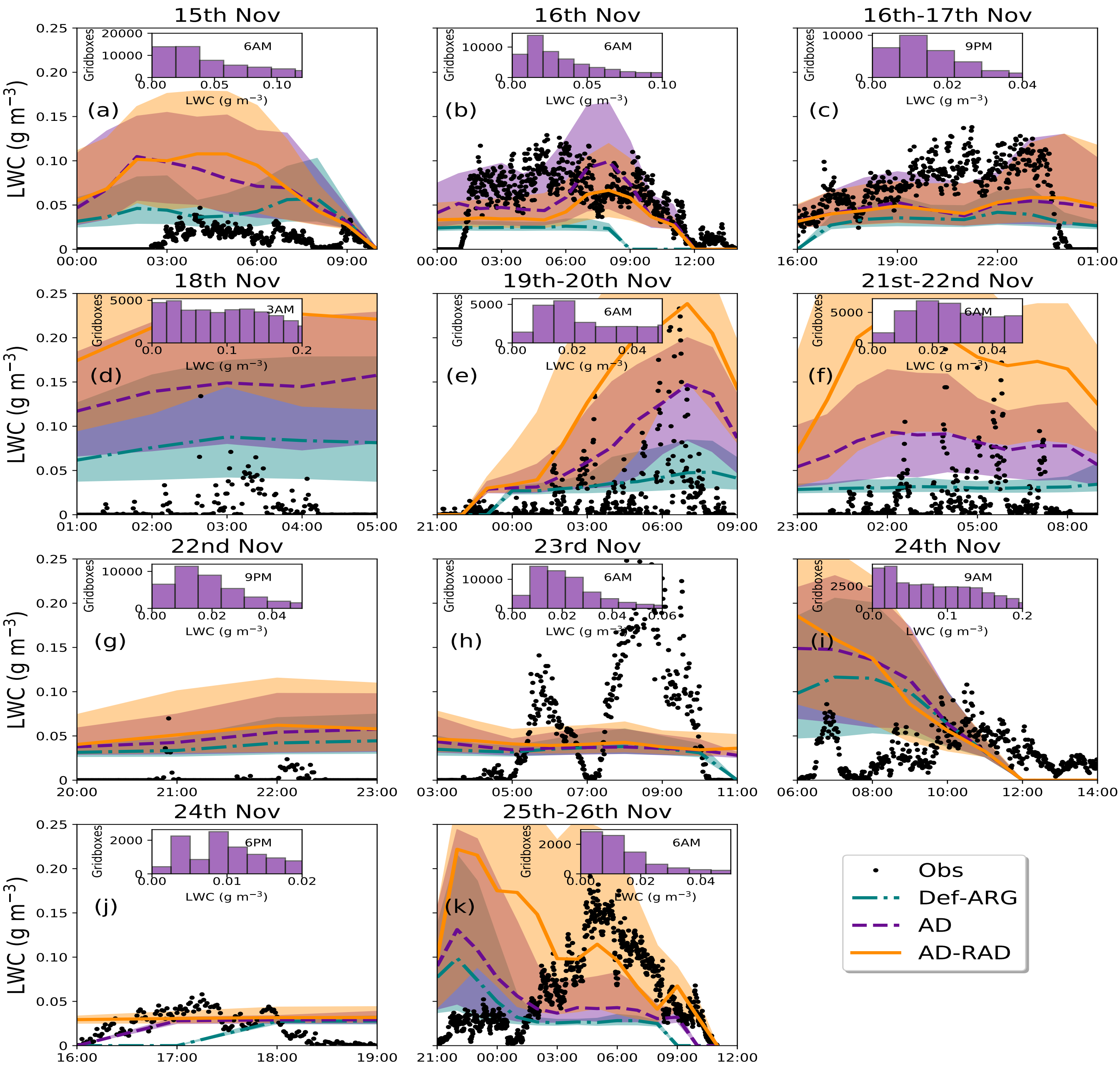

**Figure 8.** Variation of simulated and observed liquid water content as a function of time for different fog events. The results of the 500 m model at 5 m altitude from the simulations Def-ARG, AD, and AD-RAD are compared with observations at the SIRTA observatory (UTC time). The solid and dashed lines represent the median values, and the shaded regions represent the interquartile ranges over the foggy gridboxes. The inset plots show histogram of in-fog liquid water content at different times of the fog events from the AD simulation. The minimum LWC visible in the plots of around 0.025 g m$^{-3}$ is due to the thresholds for defining gridboxes as foggy, as described in the text.

25th November (subfigures a, f and k). The mechanisms for the aerosol-fog interactions we simulate are likely similar to those discussed in the companion paper.

Figure S4 shows the LWC time series for the AD-RAD-SED, AD-RAD-INV, and AD-RAD-DCAT simulations. Unlike in the case of the $N_d$ time series, AD-RAD-DCAT and AD-RAD-SED sometimes differ substantially, with AD-RAD-SED producing LWC a factor of two lower than AD-RAD-DCAT on 18th, 19th and 21st November. Including a representation of droplet interception significantly reduces the overestimate of LWC near the surface compared to AD-RAD in these three thin fogs, as well as also improving $N_d$. However, AD-RAD-DCAT is relatively similar to AD-RAD. This difference between AD-RAD-DCAT and AD-RAD-SED is expected, as increasing sedimentation removes the liquid water mass while introducing DCAT only influences the LWC via aerosol-fog interactions. Despite the improved mean LWC in simulation AD-RAD-SED compared to AD-RAD, none of the simulations substantially improved on the poor skill of simulation AD in representing LWC trends during the fogs.

## 4.4 Liquid water path

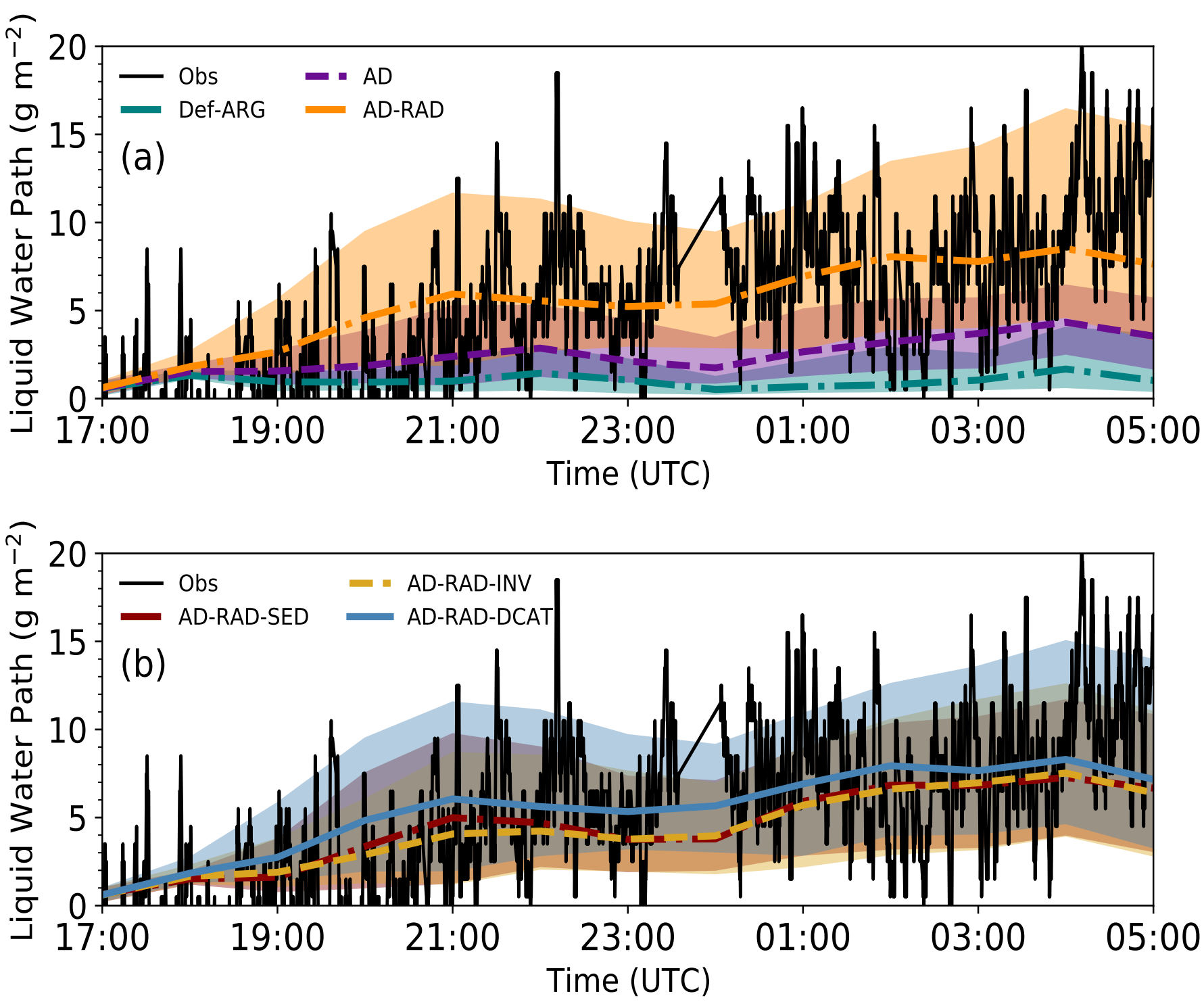

**Figure 9.** Time series of Liquid Water Path during the LANFEX case study with and without radiative cooling included in aerosol activation. Top Panel: Simulations Def-ARG, AD and AD-RAD. Bottom Panel: Simulations AD-RAD-SED, AD-RAD-INV and AD-RAD-DCAT.

We are also able to evaluate Liquid Water Path (LWP), which gives additional insight into the behavior of the fog at higher altitudes. For LANFEX, Figure 9a shows LWP time series from the microwave radiometer observations and simulations Def-ARG, AD, and AD-RAD and Figure 9b shows AD-RAD-SED, AD-RAD-INV, and AD-RAD-DCAT. To calculate LWP in simulated fog while excluding simulated clouds, we select the foggy gridboxes at the surface and calculate the LWP in those columns. As the radiometer would include LWP contributions from clouds above fog, we also allow these to contribute in the simulations. Then we calculate the median and interquartile range, and plot them as solid (and dashed) lines and shaded regions for different simulations. The Def-ARG simulation significantly underestimates the LWP during the lifetime of the fog. The AD simulation, which uses the updated ARG scheme, reduces the bias but still shows a discrepancy greater than a factor of 3 most of the time. However, despite the overestimated surface LWC we described in Section 4.3 the inclusion of radiative cooling as a source of supersaturation in the AD-RAD simulation substantially improves the model's performance, capturing the early stages of the fog within 50% of observations, but then converging to a steady mean LWP after 21:00 UTC (around 7 $\mathrm{g\,m^{-2}}$, similar to the observations). Figure 9 is similar to Figure 4(b) of Vié et al. (2024) and Figure 1 of Boutle et al. (2022). Our AD-RAD simulation is in comparably good agreement with the best performing simulations in the intercomparison. In contrast to LWC in the ParisFog cases, the trend in LWP during the fog is also in good agreement with the observations. In simulations AD-RAD-SED, AD-RAD-INV and AD-RAD-DCAT, the model performance in simulating the LWP is slightly underestimated, but better than simulation AD and similar to the DT+QC simulation of Vié et al. (2024). The underestimate mirrors the lower $N_d$ and LWC in these simulations compared to AD-RAD. In supplement Figure S5 we plot the timeseries of LWP for the ParisFog cases. The AD and AD-RAD simulations are mostly correct in determining whether fog events have low or high LWP (below or above 25 $\mathrm{g\,m^{-2}}$). However, the agreement with observations during any given fog event is mixed, and our conclusion based on LANFEX that AD-RAD performs better than Def-ARG or AD is only true in about half of the fog events.

### 4.5 Vertical Structure of Fog

We show simulated vertical profiles of $N_d$ in Figure 10 with observations from the tethered balloon during LANFEX IOP1. In our 500 m resolution model, we use the following algorithm to select grid cells to include in the vertical profile.

1. Select all foggy gridboxes at the surface level
2. To define the fog top, move up the columns that correspond to these gridboxes until fewer than 1000 still contain fog.
3. Calculate median fog properties from columns in which the surface grid box contains fog, starting at the surface and moving upwards. For example, in a hypothetical case where 20000 surface grid boxes are foggy, 15000 of these also contain fog at model level 2, 8000 of these contain fog at level 3, and 500 at level 4, the median is calculated over all of these gridboxes for levels 1-3 inclusive.

We plot the median and interquartile ranges of $N_d$ (shown by solid lines and shaded regions) at different model levels from the Def-ARG, AD and AD-RAD simulations. The top panel (subfigures a-c) show the vertical profile at different times during the LANFEX case, and other subplots show ParisFog cases.

During the LANFEX case, observations (shown in black solid and dotted lines) indicate a variable fog top height ranging from 50 m to 90 m at different times. In the model, the bottom five levels are centered at 5 m, 22 m, 45 m, 75 m, and 112 m. In simulations AD and AD-RAD, fog extends up to the 75 m level (meaning the fog top height is the top of this level, 93 m). Above this level, little or no fog is present in the model.

For this case, the Def-ARG simulation consistently underestimates $N_d$ throughout the vertical profile, though its low fog top height at 23:00 is a better match to observations than AD or AD-RAD. However, overall simulation AD seems in better agreement with the observations, although sometimes the $N_d$ is underestimated (during the mature stage of the fog, on November 25 at 03:00 UTC around 20-30 m altitude, for example). As was evident in the timeseries plots, the inclusion of radiative cooling leads to a severe overprediction of $N_d$. On average, we simulate about $100\,\mathrm{cm}^{-3}$ droplets in AD-RAD, about a factor two higher than observations but similar to Poku et al. (2021) and the LIMA simulation of Vié et al. (2024). Figure 4 of the model intercomparison study by Boutle et al. (2022) shows the vertical profile of $N_d$ simulated by several LES models. Some of these have radiative cooling included in aerosol activation, some do not. In their 'high' aerosol cases (which best matches our simulated aerosols), $N_d$ is also usually overestimated by around a factor of 2. For ParisFog, we do not have observations for vertical structure, though Mazoyer et al. (2019) comment that the fog top at SIRTA on 18, 20, 22 and 23 Nov is below 18 m altitude. Our simulations predict much higher fog top heights on these days. This apparent discrepancy may be mainly due to our selection method, which is designed to select as much fog as possible; it also seems likely from the brightness temperatures shown in Figure 3 of the companion paper that fog elsewhere in our model domain is more developed than it is at SIRTA, otherwise it likely would not be detected by the satellite. Even in simulations, the simulated fog top heights differ substantially between fog events, ranging from 80 m on 26 November to over 500 m on 23 November. The difference in fog top height between AD and AD-RAD is small on some days (e.g., the 16th, 20th, and 22nd) but significantly larger on others (e.g., the 18th, 23rd, and 26th). Similar errors frequently occur in fog simulations: for example, in the LES simulations of Boutle et al. (2022), mean fog top height is 167 m, substantially higher than observations (and our simulations).

In Supplement Figure S6, we show similar vertical profiles for the AD-RAD-SED, AD-RAD-INV, and AD-RAD-DCAT simulations. For the LANFEX case, although the vertical structure is complex, the profile simulated in AD-RAD-INV shows better agreement with observations, while the other two sensitivity studies closely resemble AD-RAD. The differences in $N_d$ at the surface are substantial across all simulations. In the ParisFog cases, the variations among sensitivity studies are much larger at all model levels.

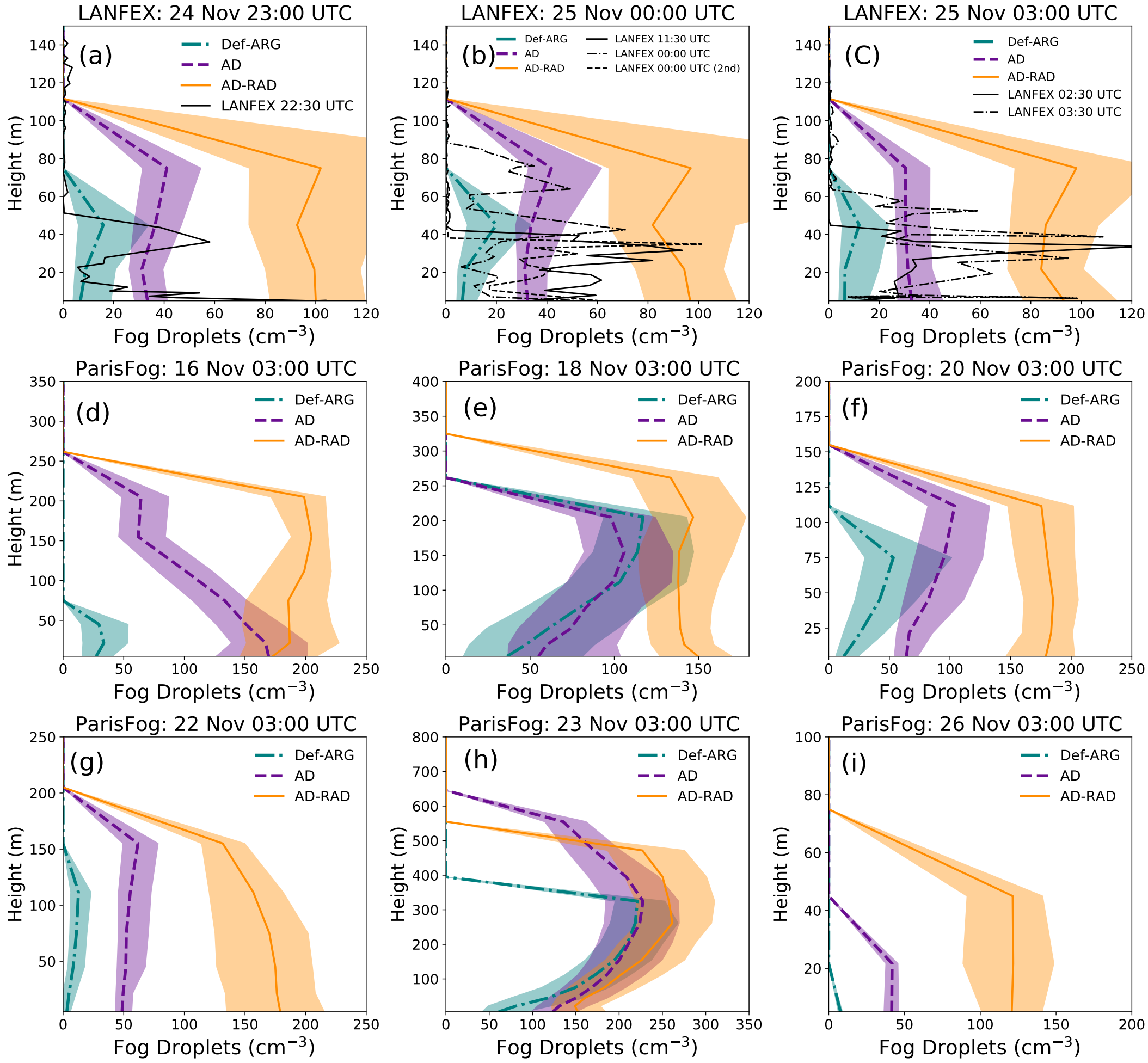

**Figure 10.** Vertical Profiles of $N_d$ from the LANFEX case study and different days of the ParisFog case study. The results of the 500 m model for Def-ARG, AD, and AD-RAD simulations are shown. Observations during LANFEX are from the fog monitor on the tethered balloon. Note the different y axis scales of the different subfigures, reflecting the substantial variability in fog top height between fog cases.

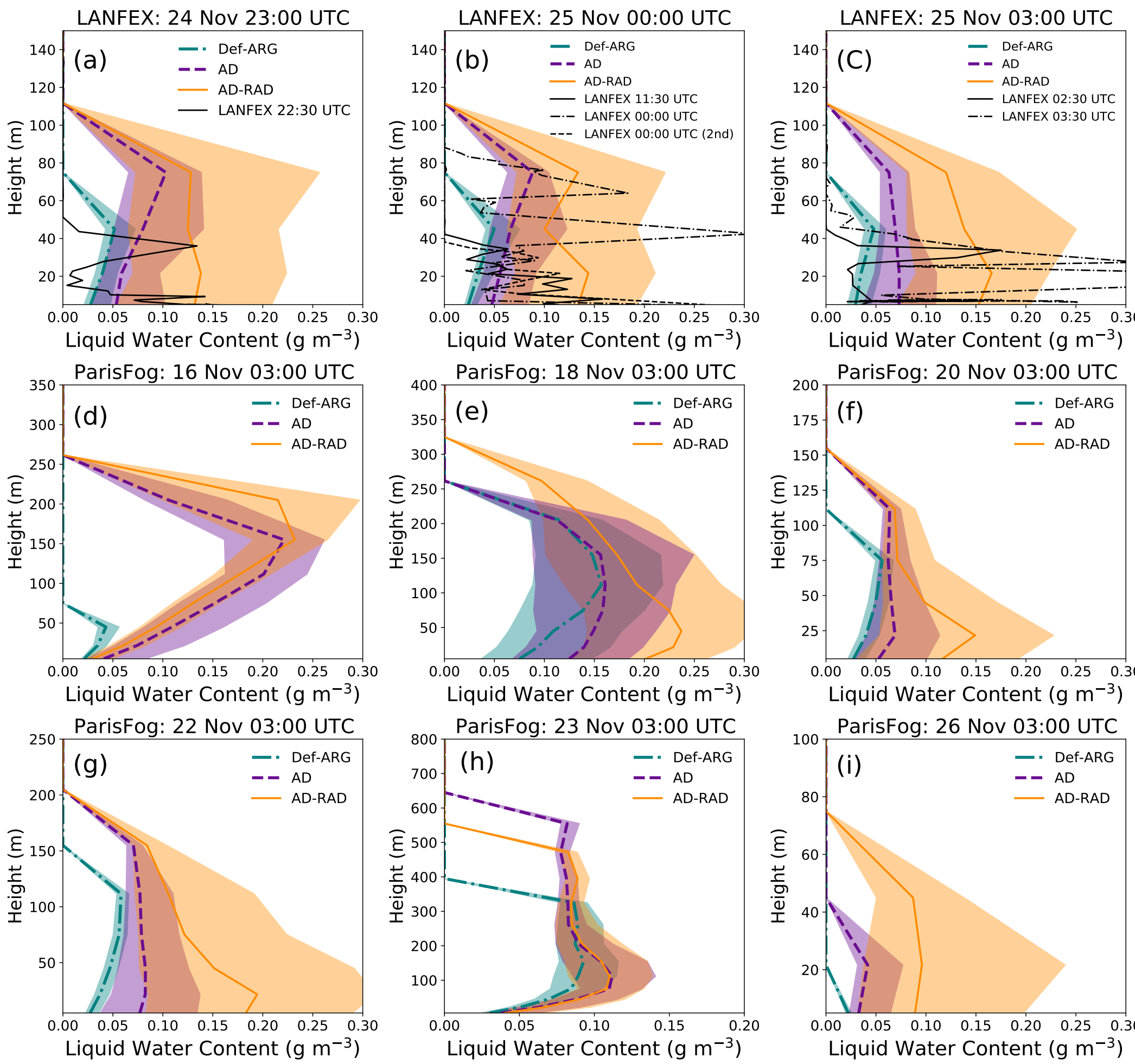

**Figure 11.** Vertical Profiles of LWC from the LANFEX case study and different days of the ParisFog case study. The results of the 500 m model for Def-ARG, AD, and AD-RAD simulations are shown here. Observations during LANFEX are from the fog monitor on the tethered balloon. Note the different y axis scales of the different subfigures, reflecting the substantial variability in fog top height between fog cases.

We show vertical profiles of LWC in Figure 11 in the 500 m-resolution model from different sensitivity simulations, similar to Figure 10. The top panel shows the LANFEX case, and other subplots are for ParisFog events. Our changes are designed to improve $N_d$ in the model but they also affect the LWC through aerosol-fog interactions (as discussed in the companion paper). Contrary to the vertical $N_d$ profiles, both simulations AD and AD-RAD are in reasonably good agreement with the LANFEX observations, but there are balloon observations (dotted black line) showing higher fluctuations of LWC near the surface and also at higher altitude. In both model and observations, $N_d$ vertical profiles are correlated to LWC vertical profiles, but the correlation is relatively weak. Hence, as also demonstrated by Vié et al. (2024), rather than assuming a perfect correlation (as done in a single-moment microphysics scheme), it is important to use the double-moment cloud microphysics scheme in the

model to accurately capture the mean behavior in the vertical profile of fog microphysical properties; however, it is likely that neither single- nor double-moment bulk microphysics can perfectly represent the variability in these profiles.

The vertical profile of LWC simulated by AD-RAD is mostly constant (except the 16th Nov ParisFog event), as it appears might be the case in the LANFEX observations if the variability were averaged out over a large area. In contrast, in simulation AD, LWC usually increases with height, qualitatively as expected in an adiabatic cloud. Since radiative cooling rates are stronger at fog top (next section), the reasons for this difference are unclear. Although $N_d$ is higher for the ParisFog cases than LANFEX, the simulated LWC in ParisFog is mostly similar to LANFEX IOP1 (about 0.15 $\mathrm{gm}^{-3}$ on average). While the Def-ARG simulation clearly simulates unrealistically low LWC, our AD and AD-RAD simulations are broadly consistent in their overall performance with the single column and LES simulations of Boutle et al. (2022); Smith et al. (2018) and Vié et al. (2024). We show similar profiles for the other simulations in Supplement Figure S7.

### 4.6 Comparison of radiative and adiabatic cooling rates

In Figure 12, we show the vertical profiles of the radiative cooling rate in the AD-RAD simulation. Figure 12a and 12b show fog events from the ParisFog case study on 15-16 November and 25-26 November 2011 respectively, while Figure 12c shows LANFEX IOP1. Other events in the ParisFog case study have similar trends. The radiative cooling rates shown in the figure are medians from all foggy grid boxes, selected in the vertical as in Figure 10. An additional colorbar shows the updraft velocities that would give cooling rates equal to the simulated radiative cooling rates, calculated assuming a temperature lapse rate of 6.5 K $\mathrm{km}^{-1}$ as in Boutle et al. (2018).

The radiative cooling rate varies with time, mostly between 0 and 1.5 K $\mathrm{hr}^{-1}$ near the surface. Equivalent updraft speeds range from $0 - 0.08$ m $\mathrm{s}^{-1}$. Once the fogs are mature, the cooling rates at higher altitude are often higher than at the surface, as expected (Boutle et al., 2018). It is possible that this could lead to activation above the base of the fog. However, this seems unlikely: Figure 10 and Supplement Figures S8 and S9, which show $N_d$ and LWC plotted in the same way as Figure 12, show generally higher droplet concentrations near the surface than at the fog top. The low equivalent updraft speeds also support the conclusion of Boutle et al. (2018) that a minimum updraft speed of 0.1 m $\mathrm{s}^{-1}$ is too high, and will artificially inflate simulated droplet concentrations in radiation fog.

In Figure 13 we show the spatial variation of the radiative cooling rate in the model at a representative time for LANFEX and two ParisFog cases. The spatial variations exceed a factor of two across our relatively small and, while not spatially homogeneous, at least not mountainous, model domains. The high variability of the radiative cooling rate suggests that using a constant minimum updraft speed is probably a poor proxy for radiative cooling in simulations of aerosol activation in fog.

To understand the relative importance of adiabatic and radiative cooling in droplet activation in these fog cases, we compared droplet concentrations in the AD-RAD simulation with those in the AD and RAD simulations. Simulation AD-RAD has both cooling terms, whereas simulation AD and RAD have only adiabatic and radiative cooling terms, respectively. Therefore the marginal contributions of the two mechanisms are as follows:

$$\text{Radiative Cooling Contribution} = \frac{N_d\,(\text{AD-RAD}) - N_d\,(\text{AD})}{N_d\,(\text{AD-RAD})}$$

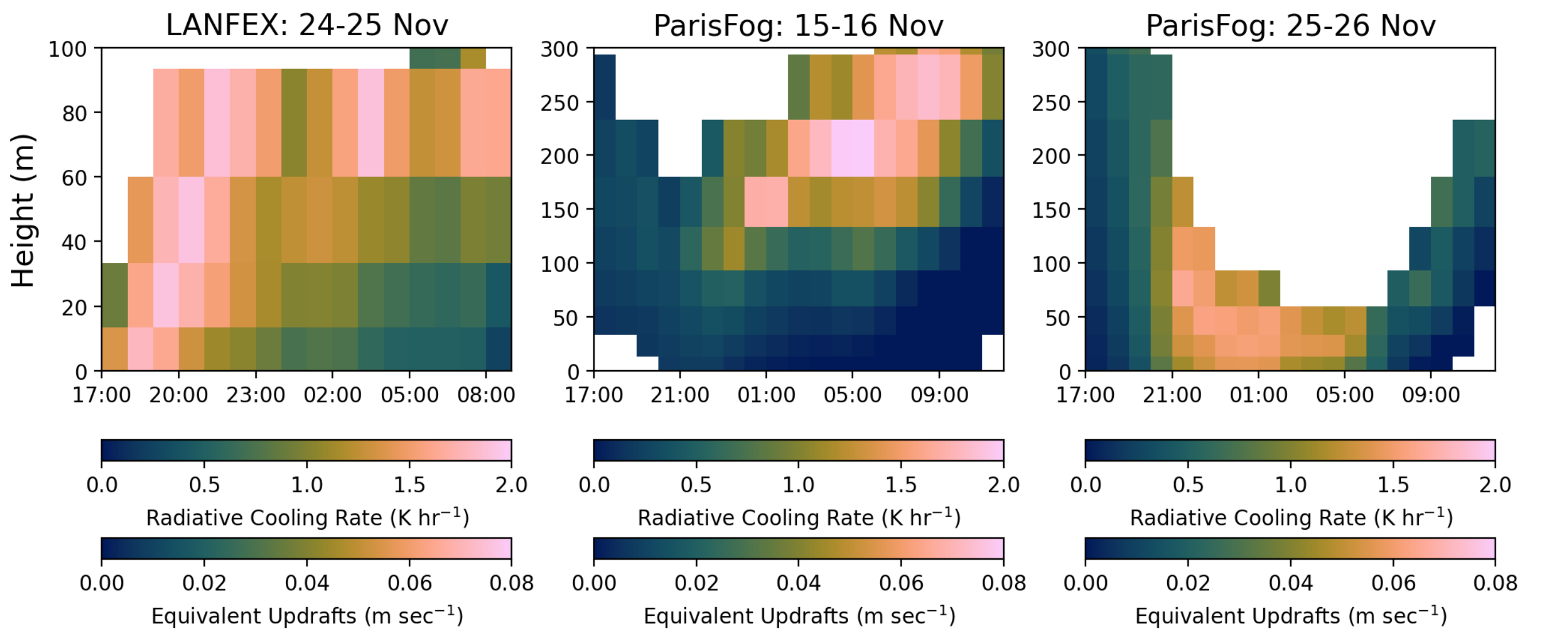

**Figure 12.** Vertical profile of radiative cooling rate in our 500 m AD-RAD simulation, as a function of time (UTC). Subfigure (a) represent the fog event from the LANFEX case, and subfigures (b), (c) represent two events from the ParisFog case. The median cooling rates of all foggy gridboxes (at the surface and their vertical column) are taken. Equivalent updrafts (assuming a 6.5 K km$^{-1}$ temperature lapse rate) are also shown in the additional colorbar.

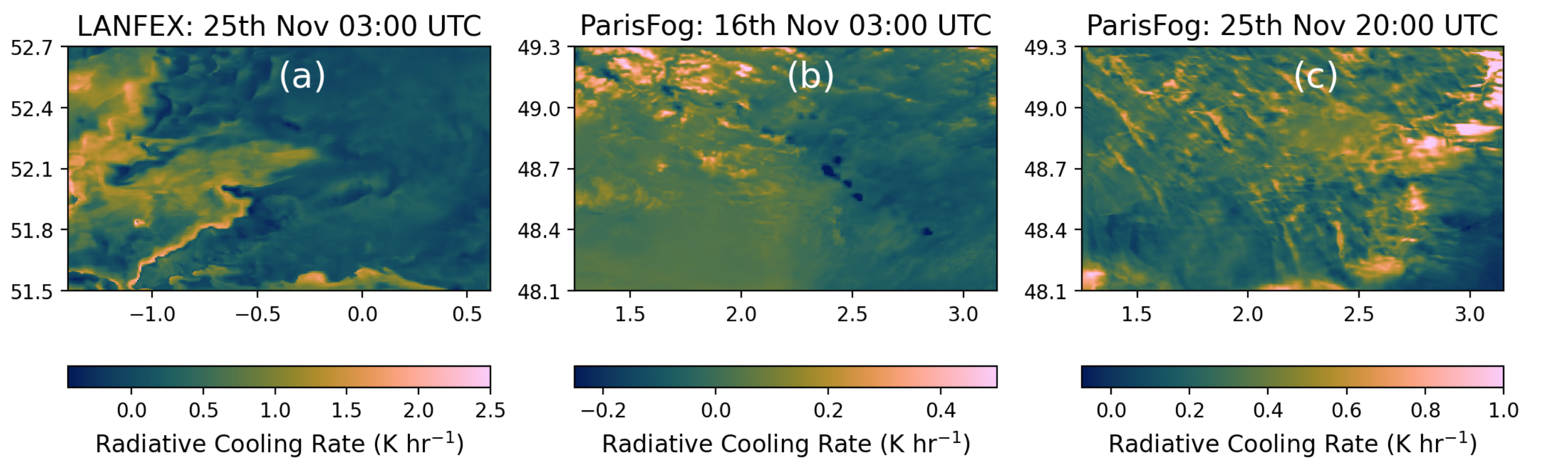

**Figure 13.** Spatial variation of radiative cooling rate at the surface from simulation AD-RAD of the 500 m model for the LANFEX case (a) and two ParisFog cases (b, c).

$$\text{Adiabatic Cooling Contribution} = \frac{N_d(\text{AD-RAD}) - N_d(\text{RAD})}{N_d(\text{AD-RAD})}$$

480 When calculating these contributions, we select the foggy grid boxes in the AD-RAD simulation and use the same locations in the other two simulations, irrespective of the presence of fog in that area. The resulting contributions are shown in Figure 14.

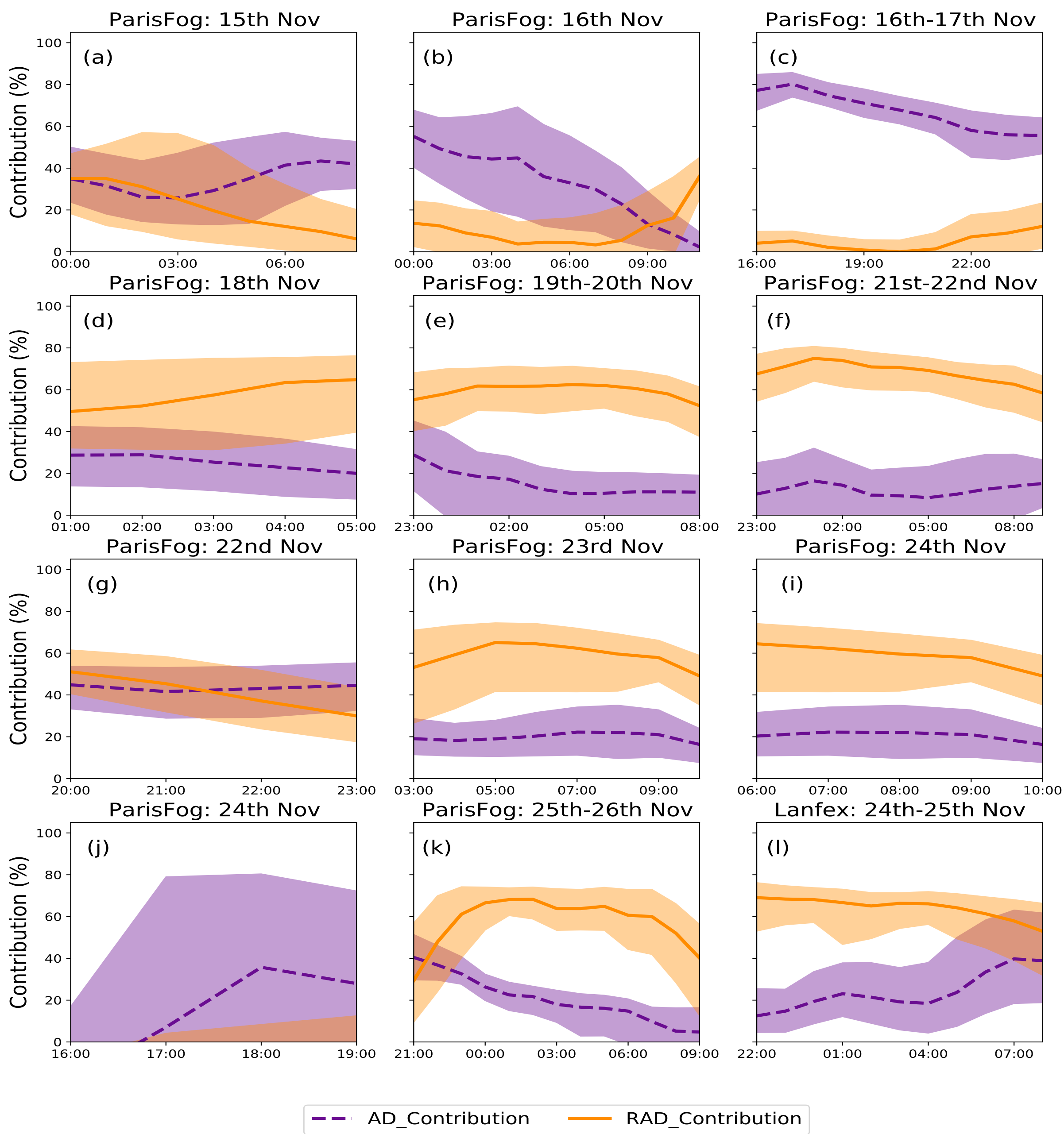

**Figure 14.** Timeseries (UTC) of the percentage contribution to droplet activation (at the surface) from radiative and adiabatic cooling, as simulated by our 500 m model across various fog cases (subplots a-k) for the ParisFog and LANFEX campaigns (subplot l). The dashed purple line represents the median fraction of droplets originating from adiabatic cooling, while the solid orange line represents those originating from radiative cooling. Shaded regions illustrate the interquartile ranges of these contributions. The AD-RAD simulation is compared with the simulations AD and RAD to generate this figure.

The two marginal contributions generally do not add to 100%, as expected, because some aerosol particles can be activated by either cooling source (so the marginal contributions sometimes amount to more than 100%), while other aerosol particles require the sum of the cooling rates to activate (so the marginal contributions sometimes sum to less than 100%).

We find the marginal contribution of radiative cooling to aerosol activation is substantial, and indeed dominant in the majority of ParisFog cases and the LANFEX case we study here. A major caveat is that results are likely influenced to some extent by the overestimation of $N_d$ in simulation AD-RAD. Qualitatively, however, this result supports the findings of Poku et al. (2021) that radiative cooling is important. In the LANFEX case, we find the contribution of radiative cooling ranges from $60-70\%$. However, for the ParisFog cases,simulated adiabatic cooling is frequently also important, even in radiation fog, and the relative contributions of radiative and adiabatic cooling have large inter- and intrafog variability for both sources. Among the ParisFog cases, on 19, 21, 23, 24 (first event) and 26 November, radiative cooling is the dominant source, with contributions ranging from $60-80\%$. During fog events on 15 Nov (starts at 03:00 UTC) and 16 November (both), adiabatic cooling is the dominant source, with a marginal contribution maximum up to 80%. For the 22 November late evening fog case, both cooling sources are equally important but vary during the fog lifecycle. In radiation fogs, we generally expect that the fog layer is initially thin and radiative cooling is dominant. With time, the fog layer thickens, and mixing within the fog layer makes the fog more adiabatic, and updraft speeds increase; so radiative cooling becomes less important (Boutle et al., 2018). In many of the fog cases (e.g. ParisFog: 15, 22, 24 Nov and the LANFEX case), but not all, we find that the contribution from radiative cooling is more important during the initial phases of fog development. As the fog layer thickens, the contribution decreases as the fog becomes more adiabatic. However, unsurprisingly, in these weather prediction simulations, the effect is harder to disentangle from case-by-case variability than in a more idealized LES simulation.

## 5 Discussion and Conclusion

In this study, we simulate radiation fog during the ParisFog and LANFEX field campaigns using 500 m resolution nested simulations with the UK Met Office Unified Model. Our approach focuses on the contribution of radiative cooling to aerosol activation. Additionally, we conduct sensitivity studies to assess the impact of other possible effects on droplet number concentrations in our cases, such as that of droplet interception or of potential biases in our simulation of boundary layer turbulence and microphysics. We evaluated our model using timeseries of surface observations and balloon measurements of fog vertical structure. Our key findings are:

- **Our model simulates sufficient fog**

  The 500 m model effectively simulates a considerable amount of fog across the domain for both the ParisFog and LANFEX cases. $N_d$ and LWC are simulated to be lower in LANFEX compared to ParisFog, which broadly agrees with our observations, and our simulated LWP in LANFEX is in excellent agreement with the observations for simulations with activation via radiative cooling included. In common with other models, our model frequently did not correctly represent the time of fog onset or dissipation, and it also had little skill in representing temporal trends in surface LWC during individual fog events.

■ **Radiative cooling is important for aerosol activation in fog**

We included a source of supersaturation from radiative cooling (together with a sink of supersaturation due to short-wave heating) into the activation scheme. With these changes, the simulated $N_d$ overestimated the observations most of the time. On the assumption that the overestimate was equally attributable to radiative and adiabatic cooling, we calculated the relative importance of radiative and adiabatic cooling in fog and found that radiative cooling is often the dominant source, but both processes are frequently important for aerosol activation, depending on the conditions. The radiative cooling rates are variable in space and time, and so cannot be properly represented by a fixed minimum updraft speed. Radiative cooling rates are always calculated by the radiation scheme in the model, independent of the activation process. Hence, including activation by radiative cooling should not significantly increase the computational cost. Similar to Poku et al. (2021), Peterka et al. (2024), and (less directly) Vié et al. (2024), our simulations highlight the possible importance of radiative cooling in the accurate representation of the droplet budget in the model.

■ **Interception by trees and buildings is probably an important droplet sink**

In our first sensitivity study, we increased the sedimentation rate at the bottom-most model level by a factor of 3 as a proxy for droplet interception. This process was not included in the model before, and would need to be generalized to different surface types (for example, ocean) before it could be applied by default. When added to our model simulations with radiative cooling included, the change reduces bias in $N_d$ and LWC. The high sensitivity of droplet concentrations to sedimentation rates corroborates the importance of fog deposition in the accurate prediction of the droplet lifecycle, as explained by Mazoyer et al. (2022).

■ **Suppression of activation via updrafts in inversions could also improve model performance**

In a second sensitivity study, we turned off activation by updrafts in temperature inversions, under the assumption that these updrafts are overestimated, leading to excessive activation, similar to the findings of Boutle et al. (2018). We found that this change improved the model performance in most of the cases. If this modification were included in the model by default together with radiative cooling in activation, the importance of radiative cooling in activation would likely be higher than the results we show in Section 4.6 suggest.

■ **Overestimates in droplet concentrations could be mitigated if the concentrations slowly adjust to changes in supersaturation over time**

In our model simulations with radiative cooling included, we find that allowing droplets to deactivate in response to reductions in supersaturation over a relatively long but adjustable timescale leads to lower concentrations that agree better with observations. We term the timescale as the Droplet Concentration Adjustment Timescale (DCAT). We tested a 10-minute timescale. The introduction of DCAT could serve as a proxy for missing processes in the model, such as droplet collision-coalescence or Ostwald ripening (Degefie et al., 2014; Mazoyer et al., 2019, 2022). While our model does not simulate the hygroscopic growth of haze aerosols accurately, as we discuss in the companion paper, if small droplets in our simulations can be considered a proxy for haze aerosols (which could also be detected as droplets by a fog

monitor), then DCAT could also represent the possibility of these droplets shrinking in response to changes in relative humidity over time.

- **Simulation of fog top height is satisfactory in LANFEX, and vertical structure could be realistic for an area average**

  In our 500 m-resolution simulations, the fog top height is simulated accurately in the model for the LANFEX case. In ParisFog, model performance may well be worse, but the comparison may be unfair as we compare measurements at a single point to an aggregate over a model domain. Simulations without radiative cooling generally simulate $N_d$ well but the LWP is underestimated during LANFEX IOP1. With radiative cooling, agreement with observations is generally worse in simulating the vertical structure of $N_d$, but in good agreement for LWC. The sensitivity studies proves the model performance. The detailed vertical structure of LWC and $N_d$ in the balloon measurements is not replicated by the model, but this is expected since these measurements are likely affected by highly localized fluctuations that are not simulated in detail by any model in the intercomparison study of Boutle et al. (2022), or in subsequent studies (e.g Vié et al., 2024).

Our two-part publication aimed to explore strategies to realistically simulate fog droplet concentrations in a weather and climate model with prognostic aerosol microphysics. The companion paperfocused on activation by adiabatic cooling during ParisFog while this paper focused on the role of radiative cooling in both ParisFog and LANFEX. We aimed to test only small modifications to the aerosol activation scheme, rather than larger changes to the model structure that would be more likely to impact the computational cost of the model or its ability to simulate low clouds. Our tests yield a better understanding of the importance of different cooling sources for fog droplet activation in non-idealized cases. However, more fog events in different environments would need to be simulated before we can confidently recommend specific developments for new model releases. The high priorities will be to simulate fog in pristine marine conditions, coastal fog, and ice fog. Furthermore, simulated updraft speeds for aerosol activation depend on horizontal grid resolution. In the simulations we present here, these updrafts are likely underestimated because we do not include a contribution from sub-grid-scale turbulence. While a scale-invariant parameterization of subgrid updraft velocity for the UM has been proposed (Malavelle et al., 2014), it still needs (at least) further testing and modification before it can be routinely used (Gordon et al., 2020). Thus, understanding how our parameterizations depend on domain resolution is an important task for the future. Finally, we also need to investigate how any changes to the activation scheme we recommend affect droplet concentrations in different cloud types. However, this work lays the foundations for improvements in aerosol activation in weather and climate models, with clear implications for the simulated optical properties and life cycle of fog.

*Code and data availability.* All model and observation data used in this work is available at: https://doi.org/10.5281/zenodo.14005081 (Ghosh et al., 2024a). All atmospheric simulations used in this work were performed using version 13.0 of the Met Office Unified Model (UM) starting from the GA7.1 configuration (Walters et al., 2019), and also included version 7.0 of JULES. The source code used in this study is free to use. However, software for this research is not publicly available due to intellectual property copyright restrictions, but is available to signatories of the Met Office Software license. Full descriptions of the software, including the specific configurations used in this study, can be found in the text of this article and in articles cited therein. A number of research organizations and national meteorological services use the UM in collaboration with the Met Office to undertake atmospheric process research, produce forecasts, develop the UM code, and build and evaluate Earth system models. To apply for a license for the UM, go to https://www.metoffice.gov.uk/research/approach/modelling-systems/unified-model (last access: Oct 23 2024; Met Office, 2024), and for permission to use JULES, go to https://jules.jchmr.org (last access: Oct 23 2024). Rose and Cylc software were used to drive the Unified Model. The simulations were run using Rose version 2019.01.3 and Cylc version 7.8.8, which are publicly available at https://doi.org/10.5281/zenodo.3800775 (Shin et al., 2020), and https://doi.org/10.5281/zenodo.4638360 (Oliver et al., 2021) respectively. Both Rose and Cylc are available under v3 of the GNU General Public License (GPL). The full list of simulation identifiers for the simulations in this paper is given below.

- ParisFog Simulation Def-ARG: u-dh438
- ParisFog Simulation AD: u-dh847
- ParisFog Simulation AD-RAD: u-cr239
- ParisFog Simulation AD-RAD-SED: u-dh855
- ParisFog Simulation AD-RAD-INV: u-cr241
- ParisFog Simulation AD-RAD-DCAT: u-cr118
- ParisFog Simulation RAD: u-cv714
- LANFEX Simulation Def-ARG: u-di008
- LANFEX Simulation AD: u-cm093
- LANFEX Simulation AD-RAD: u-cm528
- LANFEX Simulation AD-RAD-SED: u-cm530
- LANFEX Simulation AD-RAD-INV: u-cn606
- LANFEX Simulation AD-RAD-DCAT: u-dh436
- LANFEX Simulation RAD: u-dj473

*Author contributions.* PG and HG formulated the idea of the paper with important contributions from all co-authors. PG and HG designed the simulations and set up different model configurations. MM supplied the data from the ParisFog field campaign. IB supplied the data from the LANFEX field campaign. PG ran all the simulations. PG analyzed the simulation and observation data with contributions from HG, and wrote the paper with comments and suggestions from all co-authors.

*Competing interests.* The authors declare no competing interests.

*Acknowledgements.* This research was supported by the U.S. Air Force Life Cycle Management Center (LCMC) collaboration with Oak Ridge National Laboratory (ORNL). The computational resources on Air Force Weather HPC11 are provided by the Oak Ridge Leadership Computing Facility (OLCF) Director's Discretion Project NWP501. The OLCF at Oak Ridge National Laboratory (ORNL) is supported by the Office of Science of the U.S. Department of Energy under Contract No.DE-AC05-00OR22725. We thank the scientists responsible for the ParisFog and the LANFEX field campaign. Model simulations are material produced using Met Office software. This work used the Extreme Science and Engineering Discovery Environment (XSEDE), which is supported by the National Science Foundation Grant ACI-1548562. Specifically, it used the Bridges-2 system, which is supported by the NSF Award ACI-1928147, at the Pittsburgh Supercomputing Center (PSC). This work also used Bridges-2 at the PSC through allocation atm200005p from the Advanced Cyberinfrastructure Coordination Ecosystem: Services & Support (ACCESS) program, which is supported by National Science Foundation grants #2138259, #2138286, #2138307, #2137603, and #2138296.

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
