# Peer review of "Adiabatic and radiative cooling are both important causes of aerosol activation in simulated fog events in Europe"

_EGUsphere, 2024_

## Author Response (AR1)

**Response to Referee Comment 1 (Anonymous Referee #3)**

This paper is the 2nd in a pair of papers aimed at exploring the factors modulating the concentration of fog droplets (Nd) in an NWP model with prognostic aerosol microphysics. While the 1st paper focused on aerosol activation leading to fog formation via adiabatic cooling, this manuscript examined the role of radiative cooling and several additional factors which can potentially modify supersaturation and aerosol activation. They found some sensitivity to all processes that were tested, with radiative cooling sometimes being greater than adiabatic cooling. Deposition processes were also found to be important. In addition to fog droplet concentration, the authors compared liquid water content, liquid water path, and fog top height from the model to observations from the LANFEX and ParisFog field campaigns.

It is known that fog is a challenging forecast problem. The observations used here have a large degree of variability, and it is difficult to conclude that a particular model configuration performs better for the majority of cases. Nevertheless, the relative behavior of the experiments is informative and the publication of this study adds to the understanding of the model processes required to accurately simulate fog. Recommendation: Accept with minor revisions.

We sincerely thank the reviewer for their helpful and encouraging comments. Below, we provide our detailed responses, with the reviewer's comments shown in Blue and our replies in Black.

During the process of addressing the reviewer's comments, we identified a bug in our code that affected the intended modifications in the Mod-Kappa simulation in the companion paper. After fixing this issue, we observed some quantitative differences in droplet concentrations between the Mod-ARG and Mod-Kappa simulations, as shown in the companion paper. All the simulations shown in this paper include the bug fix. We have updated the figures and text accordingly to reflect the corrections. However, the key take-home messages of the manuscript have remained the same.

General Comments/Questions:

It is not surprising that radiative cooling is important in these simulations, as the fog type simulated is "radiation fog". It also seems unsurprising that the AD-RAD experiments overpredicts Nd, if the code used for AD was already modified to give improved values of Nd without radiative cooling. Why not add the radiative cooling to Def-ARG?

We agree with the reviewer that radiative cooling is indeed a key driver of supersaturation in radiation fog, so its importance in our simulations is expected. We also agree that adding radiative cooling to the Def-ARG experiment might produce droplet number predictions in better agreement with observations than adding it to AD (though this is less true now that we fixed the bug associated with hygroscopicities we mentioned above). We did not do this because it is our opinion that AD corrects deficiencies in Def-ARG, and so the model developments presented in Paper 1 (adjusting the ARG parameterization and updating hygroscopicity) should be implemented in models moving forward. The simulations we perform in Paper 2 represent sensitivity studies, and we think it premature to make a recommendation that modelers adopt a configuration in Paper 2 other than 'AD', since these model developments are more exploratory and we have only examined how the simulations perform in one location.

In theory, based on the mean of the normalized mean biases of each simulation over the different fog events (Table S1), weighting each fog event equally, the least biased simulation is AD-RAD-DCAT.

To clarify the nature of our model developments and the rationale for the split between the two papers, we modify line 97 of Paper 1 so it reads:

"In the companion paper [this study] we examine more exploratory model developments."

and in line 89 of this paper we add:

"Compared to the companion paper, these sensitivity studies generally represent explorations of possible developments to the model physics, and are not simply updating parameters."

The sensitivity of the various cloud properties to changes in the various cooling sources is shown, but is there any significant change in the spatial coverage of the fog between experiments?

We thank the reviewer for raising this concern. As shown in Figure R1 (which is copied to supplement Figure S4), the number of foggy gridboxes between simulations AD and AD-RAD does not change substantially during different fog events. Minor differences are observed on some days, such as Nov 16, 18, and 22.

[Figure]

Figure R1: Timeseries of number of foggy gridboxes at the surface of the 500 m model from simulations AD and AD-RAD during different ParisFog cases.

In line 277 of the revised manuscript, we add the following:

"Figure S4 presents the time series of the number of foggy grid boxes at the surface during the ParisFog events, as simulated by the 500 m model for both AD and AD-RAD experiments. The fog coverage is similar between the two simulations, with only minor differences observed on certain days, such as Nov 22."

The radiative effects include interaction with both the land surface and the fog, correct? It looks like your analyses are made only during foggy periods, and the droplet activation due to radiative cooling is relatively constant with time in most cases. I was curious if you were able to detect an increase in radiative effects as the fog became optically thicker? – or perhaps then the surface radiative effects would decrease?

Interesting question. Yes, the radiative effects include interactions with both land surface and with the fog; and yes, the analysis was performed only during the foggy period.

Both the radiative and adiabatic cooling rates tend to increase as the fog becomes optically thick. The relative importance of each therefore varies in a complicated way (Figure 14). However, generally the reviewer is correct that radiative effects increase as the fog becomes optically thicker.

In Figure 12 of the revised manuscript, we now show vertical profiles of radiative cooling rates for 9 fog events, including the LANFEX fog case. In Figures 10 and 11 of the revised manuscript, we now show vertical profiles of $N_d$ and LWC for 9 ParisFog cases and comparisons with observations at different times during the LANFEX case. Figures S10 and S11 show time series of these profiles for one simulation. The microphysics scheme in the model is designed for low-level clouds where most of the activation is expected to happen near the cloud base. After the onset of fog, aerosol activation may be suppressed near the fog top because as shown in Figures 11 and S11, the higher LWC at fog top acts as a condensation sink, reducing the supersaturation and suppressing activation of new droplets. Figure 12 shows that the surface radiative cooling rate generally varies substantially over time during the fogs.

In line 479 of the revised manuscript, we add:

"Cooling rates exhibit substantial variation both within and between fog cases, at the surface and at higher altitudes near the fog top. Although cooling rates are much higher near the fog top (similar to Boutle et al. (2018)), activation may be suppressed near the fog top after the time of fog onset. This is because the elevated LWC near the fog top in a well-developed fog (Figures 11, S11) can act as a condensation sink, thereby suppressing supersaturation near the fog top (Figures 12, S12)."

Specific Comments:

Line 107: "We use observations from including. . . " There seems to be something missing between "from" and "including"?

Fixed.

Top of page 10: Should the 2nd equation be subtracting the 2nd term rather than adding? If not, please clarify how "New Nd" can decrease.

No, the equation is correct. New $N_d$ can only decrease when the fog evaporates. At line 229, we say 'If the cloud fraction calculated in the model decreases, the droplets evaporate proportionally to the decrease in the cloud fraction'

How is cloud fraction computed?

In the Smith (1990) scheme, cloud fraction is diagnosed using an assumed sub-grid variability of moisture. The scheme models this variability through a symmetric triangular probability density function (PDF) of a normalized moisture variable. This allows for partial cloud cover even when the gridbox-mean relative humidity is below saturation.

The normalized moisture excess, $Q_N$, is defined as:

$$Q_N = \frac{q_T - q_{\text{sat}}(T)}{(1 - RH_{\text{crit}}) \, q_{\text{sat}}(T)}$$

where:

- $q_T$ is the gridbox total water mixing ratio,

- $q_{\text{sat}}(T)$ is the saturation mixing ratio at temperature $T$,

- $RH_{\text{crit}}$ is the critical relative humidity

The cloud fraction $C$ is then calculated analytically from the triangular PDF as:

$$C = \begin{cases} 0 & \text{if } Q_N \leq -1 \\ \frac{1}{2}(1 + Q_N)^2 & \text{if } -1 < Q_N \leq 0 \\ 1 - \frac{1}{2}(1 - Q_N)^2 & \text{if } 0 < Q_N < 1 \\ 1 & \text{if } Q_N \geq 1 \end{cases}$$

This formulation results from integrating the assumed PDF over the region where condensation occurs. It provides a smooth transition in cloud fraction from clear to overcast conditions and links the moisture variability explicitly to cloud formation.

We add to line 231, 'as calculated by the the Smith (1990) parameterization'

Fig. 14: Are there any other characteristics that would help categorize when radiative cooling is dominant?

Yes, in addition to relative changes in $N_d$ between simulations with and without activation via radiative cooling, variations in the liquid water path (LWP) can also serve as a valuable indicator of the importance of radiative cooling.

Line 561: Add a space in "paperfocused".

Fixed.

**Response to Referee Comment 2 (Anonymous Referee #1)**

Review of "Adiabatic and radiative cooling are both important causes of aerosol activation in simulated fog events in Europe" by Ghosh et al.

This study adds to a growing body of work that suggests radiative cooling is an important source of supersaturation for activation of droplets in fog. As with the companion paper, I felt that some sections could be shortened, but overall it is another straightforward comparison of simulations and observations. I have only a few comments.

We sincerely thank the reviewer for their helpful and encouraging comments. Below, we provide our detailed responses, with the reviewer's comments shown in Blue and our replies in black.

During the process of addressing the reviewer's comments, we identified a bug in our code that affected the intended modifications in the Mod-Kappa simulation in the companion paper. After fixing this issue, we observed some quantitative differences in droplet concentrations between the Mod-ARG and Mod-Kappa simulations, as shown in the companion paper. All the simulations shown in this paper include the bug fix. We have updated the figures and text accordingly to reflect the corrections. However, the key take-home messages of the manuscript have remained the same.

References should be included for the first two sentences of the introduction.

References added.

A lot of time is spent showing results for Def-ARG. Since the difference between Def-ARG and AD was already extensively covered in the companion paper, I would ask the authors to consider whether these simulation results could be mostly or entirely removed. Since not all simulations are

shown in every analysis due to clutter problems if all are shown, removing Def-ARG would allow the authors to show another simulation, perhaps AD-RAD-SED.

This is a helpful suggestion. Based on this feedback, we have shortened some of the text. However, we believe that it is important to include the default simulation for the $N_d$ and LWC timeseries, as it allows the reader to quickly grasp the model improvements presented in the companion paper. The LANFEX fog case simulation is new for this paper, and we think that it is important to show the results from Def-ARG and compare them with the results from other experiments, such that we havve more confidence in the proposed changes. Finally, we chose to retain the original time series plot to present all sensitivity experiments in one place for clarity and completeness.

Figure 5 – the legend uses "Mod-Kappa" from the companion paper rather than AD.

Thanks - Fixed.

The authors make a few comments about how the AD and AD-RAD simulations match the observed fog top well. I have a hard time seeing this. It seems like the simulated fog top is 20-50m higher than observed. Perhaps that is not a lot, but it seems substantial when the fog top is only at 50m.

In Figures 10 and 11 of the revised manuscript, we now show vertical profiles of $N_d$ and LWC for 9 ParisFog cases and comparisons with observations at different times during the LANFEX case. For LANFEX, we said in line 430 (paraphrasing) that Def-ARG had a reasonable fog top height at 2300 UTC while AD and AD-RAD had reasonable agreement at 0000 and 0300UTC: "[Def-ARG's] low fog top height at 23:00 is a better match to observations than AD or AD-RAD. However, overall simulation AD seems in better agreement with the observations."

We agree that the agreement of the fog top height with observations is not always good. The first six model levels above the surface have centers at 5 m, 22 m, 45 m, 75 m, 112 m, 155 m. So the grid spacing is coarse compared to the biases in question. Considering this coarse spacing, for the LANFEX case, the model sometimes shows good agreement with observations (even when biases exceed 20 m, this is within one model level if the fog top exceeds 80 m altitude) but sometimes does not.

The biases are accentuated by our plotting the first model level at which $N_d$ goes to zero in Figure 10 and 11. We changed the figure to avoid this. We added a more detailed commentary on the model's behavior that sets it in the context of the vertical resolution at line 432:

"the fog top height matches locations where $N_d > 20\,\mathrm{cm}^{-3}$ and LWC (Figure 11) is greater than $0.05\,\mathrm{g m}^{-3}$ within one model level. However, Figure 11 suggests the fog top is one level too high in both AD and AD-RAD at 03:00 UTC"

The definition of the simulated fog top height as the highest model level in which at least 1000 gridboxes are foggy is also somewhat subjective, and a different definition might lead to different levels of agreement with observations. See our response to 'Referee Comment 3' for a more detailed discussion.

**Response to Referee Comment 3 (Anonymous Referee #2)**

The article "Adiabatic and radiative cooling are both important causes of aerosol activation in simulated fog events in Europe" by Ghosh et al. addresses an important scientific question about "aerosol-cloud" interactions, which remains a major uncertainty in estimates of the indirect forcing exerted by aerosols.

In this paper, author's explore the contribution of radiative cooling to activation for different fog cases and examine a single case study from the local and Non-Local Fog Experiment (LANFEX) field campaign in 2014. This study includes different sensitivity simulations which are clearly presented in the document. These tests mainly focus on the interception of droplets by trees, updrafts in temperature inversions and potential mechanisms for droplet deactivation. The paper is well written with a complete presentation of the various simulations as well as the comparisons with different in-situ observations of fog. This article should be published after some modifications. What is missing at this stage is a more detailed analysis of the various biases presented in this study. This needs to be improved.

We sincerely thank the reviewer for their helpful and encouraging comments. Below, we provide our detailed responses, with the reviewer's comments shown in Blue and our replies in Black. Two other reviewers suggested shortening the text, while you have requested further clarification on several aspects. To address the feedback from all reviewers, we have provided additional clarifications in the responses below, while keeping the added text minimal in the revised manuscript.

During the process of addressing the reviewer's comments, we identified a bug in our code that affected the intended modifications in the Mod-Kappa simulation in the companion paper. After fixing this issue, we observed some quantitative differences in droplet concentrations between the Mod-ARG and Mod-Kappa simulations, as shown in the companion paper. All the simulations shown in this paper include the bug fix. We have updated the figures and text accordingly to reflect the corrections. However, the key take-home messages of the manuscript have remained the same.

Regarding the aerosol and fog droplet size distribution for 16 November, Figure 5 shows that the AD simulation is actually better than the Def-ARG simulation for diameters around 10 µm, but above 30 µm it seems to be the opposite. Is there any explanation for this?

The reviewer raises an insightful point. From Def-ARG to AD, Figure 6b shows that $N_d$ increases by a factor of five while LWC increases by a factor of 2 (Figure 8b). Because $N_d$ increases much more than LWC, the number increases between Def-ARG and AD while the diameter slightly decreases. Overall, AD is a better match to both $N_d$ and LWC (Figures 6b and 8b), even though Def-ARG predicts a droplet diameter in slightly better agreement with observations. This is because the simulated LWC is biased slightly low, for reasons that may not relate to aerosol-fog interactions (for example, Paper 1 Figure 5 suggests the inversion height on 16th November is too high).

This pattern is not consistently observed in all fog cases. In our model, $N_d$ is represented using a gamma distribution, which may not always accurately capture the observed droplet size distribution. Furthermore, observational measurements are point-based and can reflect highly localized variability. Therefore, direct comparisons between spatially averaged modeled $N_d$ distributions and point-based observations can be imprecise. However, given the limited availability of observational data, such comparisons remain one of the most viable approaches. It is also important to acknowledge the inherent limitations of the measurement instrument, which may further contribute to discrepancies.

In line 303 of the revised manuscript, we add the following: "On November 16, the observed droplet size distribution is in better agreement with simulation AD and AD-RAD around 10 µm, however above 30 µm, simulation Def-ARG seems to perform better. However, we show later (Figure 6b) that $N_d$ on this day is in much better agreement in AD and AD-RAD compared to Def-ARG, despite Def-ARG predicting droplet diameters that are slightly closer to observations. This discrepancy arises mainly because Def-ARG underestimates the LWC, and this LWC bias shifts the gamma distribution used in the model towards larger droplet diameters."

For the 25 November (LANFEX), the AD simulation clearly improves the comparisons for fog droplets but there is still a significant underestimation of concentrations. It would be interesting to

discuss more such biases in this section. For this figure, it would also be interesting to add at least the AD-RAD simulation (as in part 4.3) for the aerosol and droplet size distributions and discuss the possible differences.

We thank the reviewer for this great suggestion. We have updated the plot to include the $N_d$ distribution from the AD-RAD simulation (shown in a solid orange line). Please, see the updated section 4.2. We show that for the LANFEX fog case, simulation AD-RAD is in better agreement with the observations, while for the ParisFog case, the changes to droplet size distributions are minimal. Furthermore, we show that the AD-RAD simulation overestimates the vertical profile of $N_d$ during the LANFEX case. Near the surface, the measurements exhibit large variability, which may contribute to an apparent underestimation of droplet concentrations at the surface, even in the AD-RAD simulation.

In line 297 of the revised manuscript, we add:

"At 03:00 UTC during the ParisFog case, the droplet size distribution in the AD-RAD simulation changes only marginally compared to the AD simulation. In contrast, for the LANFEX case, $N_d$ is higher in AD-RAD than in AD, and the distribution in AD-RAD shows better agreement with observations, though it still underestimates them. The comparison is discussed further in Section 4.3. The increase in $N_d$ in the AD-RAD simulation compared to AD suggests that radiative cooling plays a more significant role in fog development in the LANFEX case."

Concerning the timeseries of fog droplet concentration and for the fog case of 16 November, the results indicate that the radiative cooling is shown to be unimportant and that two simulations, AD and AD-RAD, behave similarly. Is there any explanation for this? For the same case (16 November, Figure 6b) after 09:00, the AD simulation follows the droplet concentration variation perfectly, in contrast to the AD-RAD simulation (which shows an important overestimation after 09:00). It would be interesting to investigate the underlying processes in this case.

We thank the reviewer for raising this point. We tried to investigate this further. For the 16 November case, radiative cooling rates near the surface on this day are lower than in other fog events, at around 0.2 K hr$^{-1}$ (Figure 12), corresponding to an equivalent updraft of 0.01 m s$^{-1}$. In contrast, during the 26 November fog event (where radiative cooling is the dominant contributor, as shown in Figure 14k), the cooling rate is substantially higher, around 1.5 K hr$^{-1}$ (equivalent updraft of 0.06 m s$^{-1}$). Since we think most of the activation after the initial onset of the fog happens below the fog top on 16 November, at least in our simulations, the contribution of radiative cooling towards droplet activation is small and therefore simulations AD and AD-RAD are similar for much of the time (Kindly see replies to your other comments towards the end of this document).

In Figure R2, we show the vertical profiles of $N_d$, droplet sedimentation velocity, LWC, updrafts and radiative cooling rates at three different times during the 16 November ParisFog case. However, although radiative cooling rates at the surface remain low on 16 November, they do reach approximately 2 K hr$^{-1}$, corresponding to an equivalent updraft of 0.08 m s$^{-1}$, near the fog top. While this enhanced cooling at the fog top is unlikely to lead to much droplet activation there due to the high LWC at the same altitude, even a small increase in activation at lower altitudes seems to lead to significantly higher $N_d$ and LWC throughout the fog due to the feedback through reduced losses of droplets due to reduced sedimentation velocities, shown in Figure R2. This may also affect, or be partly responsible for, the high fog-top radiative cooling rates (Figure 12 shows the AD-RAD simulation), and these in turn may influence the development of the fog water content. However, the effects at the surface, where we assume almost all activation occurs, are very subtle, and we cannot be sure this explanation is correct. We unfortunately do not have model diagnostics that could tell us exactly at which altitude most of the activation is occurring. We did not find any differences between the simulated updrafts in AD and AD-RAD at different times.

[Figure]

Figure R2: Vertical profiles of $N_d$, droplet sedimentation velocity, LWC, updraft velocity and radiative cooling rates in fog for the 16th November ParisFog case for simulations AD and AD-RAD.

In line 513 of the revised manuscript, we add the following:

"This is most likely due to substantially lower radiative cooling rates below the fog top during these fog events compared to other cases (see Figure 12). The radiative cooling rate at the surface is approximately 0.2 K hr$^{-1}$, corresponding to an equivalent updraft of about 0.01 m s$^{-1}$. In contrast, during the 26 November fog event (where radiative cooling is the dominant driver of aerosol activation) the cooling rate is significantly higher, around 1.5 K hr$^{-1}$ (equivalent updraft of 0.06 m s$^{-1}$)."

In line 520 of the revised manuscript, we add the following:

"On 16 November, after 06:00 UTC, the fog top radiative cooling rates increase compared to those at fog onset, perhaps enhancing droplet activation at lower altitudes and leading to increased droplet concentrations near the surface, as shown in Figure 6b."

Finally, at the end of this section, it is rather difficult to get an idea of the "best" model configuration between AD and AD-RAD or other simulations. It would be interesting to summarize the results of all the comparisons presented in Figures 6 and 7 in a table with the NMBF and the NMEF for the different days studied.

In response to the reviewer's comments, we have presented the NMBF and NMEF values in the Supplement tables S1 and S2. In line 384 of the revised manuscript, we add the following summary paragraph:

"In summary, we find that incorporating radiative cooling as a source of supersaturation leads to increased surface droplet concentrations across all fog events. NMBF and NMEF are presented for all fog cases in Supplement Tables S1 and S2. On certain days, such as 16 November, the relative increase in $N_d$ is minimal at this altitude for the majority of the fog, highlighting the dominant role of adiabatic cooling. In contrast, on days like 26 November, we find a substantial increase in $N_d$, indicating a stronger contribution from radiative cooling. Sensitivity studies generally result in a reduction in $N_d$, bringing the model outputs into better agreement with observations. Consistent with the findings of the companion paper, the LWC shows a lower sensitivity to changes in aerosol activation than $N_d$, and the model exhibits limited skill in capturing the time variation of observed surface LWC across different simulation configurations."

Concerning the Nd vertical profiles, the results indicate important bias in the AD-RAD simulations compared to observations. In addition, the difference in fog top height between AD and AD-RAD is very large for some days (18 or 23 Nov). Could this point be discussed further? What are the possible causes for such differences? Perhaps it would be interesting to plot the vertical profiles of relative humidity or temperature for both simulations.

[Figure]

Figure R3: Vertical profiles of number of foggy gridboxes for the 16[th], 18[th], and 23[rd] November ParisFog cases (at 03:00 UTC) for simulations AD and AD-RAD. Red and blue dotted vertical lines represent 1000 (our threshold) and 6760 (10% of 260 x 260 gridboxes used in the analysis) foggy gridboxes, respectively. The 'Height' axis is different for the 23 November fog case.

We thank the reviewer for raising this point. With the bug related to hygroscopicities in simulation 'AD' fixed (see replies to comments in companion paper), the vertical profile of $N_d$ in simulation AD-RAD is now in better agreement with the observations during the LANFEX fog case. There are still discrepancies between the model and observations, however, and these are discussed in more detail in our replies to Referee Comment #2.

In Figures 10 and 11 of the revised manuscript, we now show vertical profiles of $N_d$ and LWC for 9 ParisFog cases and comparisons with observations at different times during the LANFEX case. For the ParisFog case, we agree with the reviewer that there also appears to be a difference in fog top height between AD and AD-RAD on both November 18 and 23. However, defining the fog top is not straightforward and using different thresholds can lead to variations in the fog top height.

In Figure R3 (copied as supplementary Figure S7), we show the number of foggy gridboxes as a function of height for three fog cases: November 16, 18, and 23 at 03:00 UTC. Red and blue dotted vertical lines represent 1000 (our threshold) and 6760 (10% of 260 x 260 gridboxes used in the analysis) foggy gridboxes, respectively. For the November 18 and 23 cases, the differences in fog top height arise due to the threshold of 1000 gridboxes we use in this paper. If we instead apply a threshold of at least 10% foggy grid boxes (6760), the fog top height becomes consistent between AD

and AD-RAD: 150 m for November 18 and 280 m for November 23. However, applying the same threshold to the November 16 case results in different fog top heights: 150 m for AD and 200 m for AD-RAD. Therefore, we believe that any differences (mostly one model level) in fog top height between AD and AD-RAD are primarily due to the choice of threshold. Kindly see the companion paper for vertical profiles of temperature and relative humidity. We did not find any differences in the T and RH profiles between different simulations (not shown).

In Line 448 of the revised manuscript we add the following:

"However, defining a fog top height is not straightforward. In Supplement Figure S7, we show the vertical profile of number of foggy gridboxes from simulations AD and AD-RAD at 03:00 UTC on 16 Nov, 18 Nov and 23 Nov. For the November 18 and 23 cases, the differences in fog top height arise due to the threshold of 1000 gridboxes we use in this paper. If we instead apply a threshold of at least 10% foggy grid boxes (6760), the fog top height becomes consistent between AD and AD-RAD: 150 m for November 18 and 280 m for November 23. However, applying the same threshold to the November 16 case results in different fog top heights: 150 m for AD and 200 m for AD-RAD. Therefore, we believe that any differences (mostly one model level) in fog top height between AD and AD-RAD are primarily due to the choice of threshold."

For the percentage contribution to droplet activation from radiative and adiabatic cooling, the different results indicate that for 19, 21, 23, 24 and 26 November, radiative cooling is the dominant source. However, during the fog events of 15 and 16 November, the results show the opposite and the adiabatic cooling is the dominant source. What could explain this important contribution for these two particular days?

We agree that on 15 and 16 November, adiabatic cooling contributes more significantly to droplet activation compared to fog events from 19-26 November. Figure 12 of the revised manuscript now shows the vertical profile of radiative cooling in many more fog events than in the submitted preprint. On 15 and 16 November, this figure shows that the radiative cooling near the surface is significantly lower than on other days. The peak radiative cooling rate is similar or even higher, but that high radiative cooling occurs only at higher altitudes. This is probably because the inversion height on 15 and 16 November is higher than on most other days (Paper 1 Figure 5). Since we think activation on 15 and 16 November well after the fog onset occurs below the fog top, because LWC is highest at fog top (Figures 11, S11) which will suppress supersaturation, radiative cooling plays a greater role in activation on 19-26 November than on 15 and 16.

In line 512 of the main text, we added the following in response to an earlier comment: "This is most likely due to substantially lower radiative cooling rates near the surface during these fog events compared to other cases (see Figure 12). The radiative cooling rate at the surface is approximately $0.2$ K hr$^{-1}$, corresponding to an equivalent updraft of about $0.01$ m s$^{-1}$. In contrast, during the 26 November fog event (where radiative cooling is the dominant driver of aerosol activation) the cooling rate is significantly higher, around $1.5$ K hr$^{-1}$ (equivalent updraft of $0.06$ m s$^{-1}$). "

We continue this in line 516: "Although the peak cooling rates on 15 and 16 November are higher and occur near the fog top (likely due to a slightly elevated inversion height, as shown in Figure 5 of the companion paper), it is possible that most droplet activation during these cases (after the initial fog onset) still occurs below the fog top, where the liquid water content (LWC) is lower (Figures 11, S11). This could explain why radiative cooling plays a less significant role in droplet activation during the 15 and 16 November fog events."

Minor points:

In Table 1 and for the AD simulation, the contribution of radiative cooling is not taken into account.

If this is well the case, Rad Cool should be indicated as "No" in the table for this simulation.

Fixed

On line 294, the authors indicate that "For simulation AD-RAD, similar to the ParisFog case, the droplet concentrations at the surface are in better agreement with the observations". Is this comparison shown in the article ?

We have removed the sentence and updated the text based on the reviewer's earlier suggestion.

**References**

Boutle, I., Price, J., Kudzotsa, I., Kokkola, H., and Romakkaniemi, S. (2018). Aerosol–fog interaction and the transition to well-mixed radiation fog. Atmospheric Chemistry and Physics, 18(11):7827–7840.

Smith, R. N. B. (1990). A scheme for predicting layer clouds and their water content in a general circulation model. Quarterly Journal of the Royal Meteorological Society, 116(492):435–460.